# Optogenetic storage and release of protein and mRNA in live cells and animals

Chaeyeon Lee[1], Jeonghye Yu [1], Jongpil Shin [1], Jihwan Yu [1], Yungyeong Heo [1], Moosung Lee[2,3], Daseuli Yu [4] ✉, YongKeun Park [2,3,5] ✉ & Won Do Heo [1,6,7] ✉

Cells compartmentalize biomolecules in membraneless structures called biomolecular condensates. While their roles in regulating cellular processes are increasingly understood, tools for their synthetic manipulation remain limited. Here, we introduce RELISR (Reversible Light-Induced Store and Release), an optogenetic condensate system that enables reversible storage and release of proteins or mRNAs. RELISR integrates multivalent scaffolds, optogenetic switches, and cargo-binding domains to trap cargo in the dark and release it upon blue-light exposure. We demonstrate its utility in primary neurons and show that light-triggered release of signaling proteins can modulate fibroblast morphology. Furthermore, light-induced release of cargo mRNA results in protein translation both in vitro and in live mice. RELISR thus provides a versatile platform for spatiotemporal control of protein activity and mRNA translation in complex biological systems, with broad potential for research and therapeutic applications.

Cells efficiently organize their internal spaces to regulate various biochemical reactions by using both membrane-bound organelles and membrane-less biomolecular condensates[1]. Despite lacking membranes, biomolecular condensates create separated compartments through multivalent interactions of scaffolds[2]. Scaffolds can also phase separate with other proteins, known as clients, that cannot phase separate alone[3]. The interaction between scaffolds and clients influences the kinetics and structure of phase separation, encoding specific functions of biomolecular condensates[4]. Biomolecular condensates serve as reaction hotspots that facilitate efficient cellular processes (e.g., signalosomes, nucleolus) and as storage sites where biomolecules are protected from interacting with external factors (e.g., RNP granules, stress granules)[1,5].

Researchers have sought to leverage these functions by developing artificial condensates. Previous approaches primarily focused on creating hotspots for cellular processes or mimicking the physical properties of natural condensates using scaffolds with low complexity regions (LCRs)[6-9]. While these tools achieved their intended goals of enhancing specific signaling pathways or replicating droplet-like behavior, they were not designed to sequester specific biomolecules. Instead, they interact dynamically with various biomolecules, lacking the ability to enable precise target sequestration. Moreover, the existing synthetic condensate platforms for protein storage[10] are not capable of storing mRNA.

We thus set out to construct an artificial condensate that can store and precisely release both proteins and mRNA. Here, we describe our developed platform, named RELISR (Reversible Light-Induced Store and Release), which can store proteins and mRNA and release them in a light-dependent manner in vitro and in vivo (Fig. 1a).

We develop RELISR by fusing a valency-optimized multivalent protein and a cargo-binding domain to an optogenetic switch, achieving efficient storage and release of targeted proteins and mRNA. We demonstrate the versatility and robustness of RELISR in various cellular contexts, including highly compartmentalized cells such as

[1]Department of Biological Sciences, Korea Advanced Institute of Science and Technology (KAIST), Daejeon, Republic of Korea. [2]Department of Physics, Korea Advanced Institute of Science and Technology (KAIST), Daejeon, South Korea. [3]KAIST Institute for Health Science and Technology, KAIST, Daejeon, South Korea. [4]Life Science Research Institute, KAIST, Daejeon, Republic of Korea. [5]Tomocube Inc, Daejeon, South Korea. [6]Department of Brain & Cognitive Sciences, KAIST, Daejeon, Republic of Korea. [7]KAIST Institute for the BioCentury (KIB), KAIST, Daejeon, Republic of Korea. ✉e-mail: sureeyu@kaist.ac.kr; yk.park@kaist.ac.kr; wondo@kaist.ac.kr

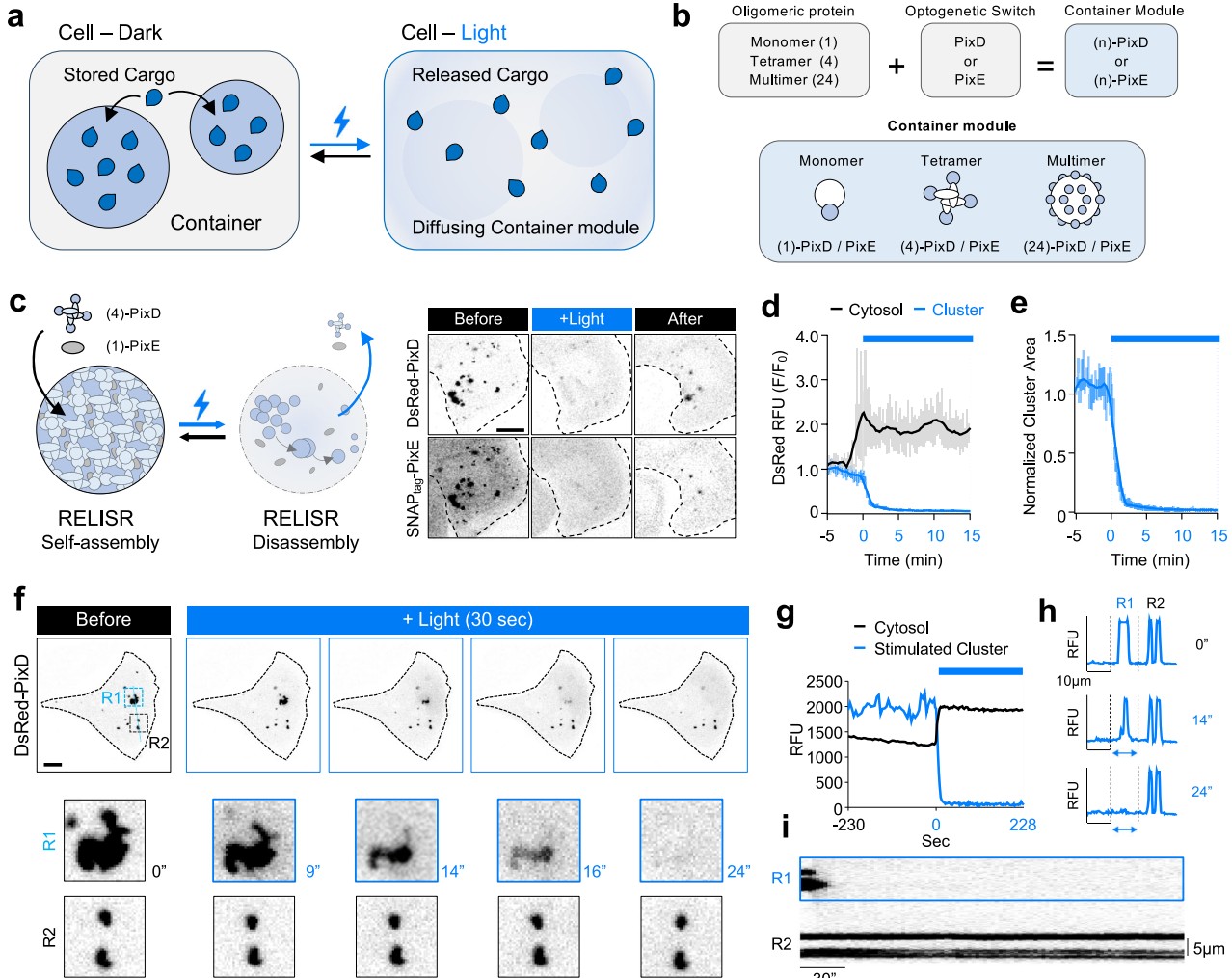

**Fig. 1 | RELISR constitutes a light-dissociable container using multivalent proteins. a** Overview of the RELISR system. Illustration of cargo storage in the dark state and light-induced cargo release. **b** Design rationale of constructing container modules. Multivalent proteins are conjugated to the optogenetic switch PixD or its binding partner PixE. **c** Schematic illustration and fluorescence images of the selected RELISR system. DsRed(4)-PixD and SNAPtag(1)-PixE form clusters in the dark state. Upon light stimulation for 15 min with 2.5 μW/cm² laser intensity, clusters disassembled. "After" images were obtained after an hour from light stimulation ends. Confocal fluorescence images of each component in inverted monochrome, expressed in HeLa cells. Black dotted line shows the outline of the cell. Scale bar, 10 μm. **d** Time-lapse graph of (**c**). Before (−5 min) and during light stimulation from 0 min to 15 min. Blue box indicates the light session. Black line (cytosol intensity) and blue bold line (cluster intensity) show the means, and gray line and light blue line show SEM of each group. **e** Time-lapse graph of (**c**), showing normalized cluster area. The bold blue line represents the mean area fraction, and the light blue shading indicates the standard error of the mean (s.e.m.). **f** Local stimulation of the RELISR system. Local stimulation started at 0 s and continued for 228 s with laser intensity 2.0 μW/cm². The images show only during 30 s of illumination. R1 indicates the local stimulation region, and R2 indicates the non-stimulated region. Blue dotted line indicates where intensity profiles were analyzed. Scake bar, 10 μm. This is a representative image from at least three independent experiments with similar results. **g** Time-lapse graph of experiments depicted in (**f**). The black line represents the cytosol, and the blue line represents the stimulated ROI (R1). Blue box indicates the light session. **h** Intensity profile for R1 and R2. Blue line indicates the fluorescence signal of DsRed-PixD. R1 signal depicted between two black dotted lines. **i** Kymograph of DsRed-PixD, including area from R1 to R2. The blue box indicates the R1 area. Scale bar, 30 s and 5 μm.

primary hippocampal neurons, and describe its use to control RhoGTPases and GEFs to achieve light-dependent morphological changes in cells. Furthermore, RELISR successfully modulated the translation of cargo mRNA in vitro and in mice, highlighting its potential to be broadly applied in biological research and therapeutic approaches.

## Results

### Design of the optogenetic container system

Multivalent interactions play pivotal roles in forming biomolecular condensates within cells. These interactions enable scaffolds—essential building blocks of biomolecular condensates with multiple binding sites, such as intrinsically disordered regions (IDRs), short linear motifs, and nucleic acids—to facilitate condensate formation in the cytosol[11].

We established our design rationale to build a light-dissociable condensate by combining multivalent proteins and optogenetic switches as scaffolds (Fig. 1a, b). To achieve this, we employed the multivalent optogenetic switch PixD (Slr1694; ~17 kDa)[12], along with its binding partner PixE(~42.3 kDa). PixD serves as the photoactive component and contain a flavin adenine dinucleotide (FAD) chromophore, functioning as a blue-light using FAD (BLUF) photoreceptors[13,14]. In the dark, PixD and PixE form a super-complex(PixD₁₀-PixE₄)[15] through hetero-oligomerization, which dissociates upon blue light illumination. Although PixD/PixE is less commonly used than other optogenetic systems that mediate light-induced association, it possesses the

unique capacity to support both multivalency and dissociation, which are critical for reversible condensate dynamics[16]. These features made the PixD/PixE pair particularly well-suited for constructing a light-dissociable condensate system, enabling us to control storage and release behavior in a spatiotemporally precise manner. We initially fused two monomeric proteins to PixD and PixE to generate mScarlet-PixE and mCerulean-PixD, respectively; however, their co-expression did not form any visible cluster (Supplementary Fig. 1a).

To amplify the multivalent characteristics of the super-complex, we attached oligomeric proteins to the N termini of PixD and PixE to build what we called "container modules" that can assemble to sequester a cargo (Fig. 1b). We tested the clustering of all combinations of PixD and PixE conjugated to the following: the fluorescent monomeric proteins (marked as '1' in the naming convention used below; used for visualization), mScarlet, mCerulean, or SNAP$_{tag}$ (a small peptide for labeling proteins with a synthetic probe[17]); a fluorescent tetrameric module (marked as '4'; used to amplify clustering and enable visualization), such as DsRed or LanYFP; and mScarlet- or SNAP$_{tag}$-tagged ferritin light chain (FTL, marked as '24'; to further amplify clustering and enable visualization), which forms a globular complex of 24 ferritin subunits[18]. We co-expressed these container modules in every possible combination (Supplementary Fig.1a).

To determine the most optimal framework among combinations, we established the evaluation criteria: the ratio of cells with clusters to the total cells ("Cluster formation rate", Supplementary Fig. 1b), how well clusters disassemble in response to light ("Light-responsiveness", Supplementary Fig. 1c), the size of individual clusters ("Single cluster size", Supplementary Fig. 1d), and the number of clusters per cell (Supplementary Fig. 1e).

Interestingly, the cluster formation rate (Supplementary Fig. 1b) was distinctly low when PixD was fused to a monomeric protein: mScarlet-PixE/mCerulean-PixD (1, 1), DsRed-PixE/mCerulean-PixD (4, 1), and FTL-SNAP$_{tag}$-PixE/mCerulean-PixD (24, 1) had cluster formation rates of < 20%, whereas the other six combinations formed clusters in over 70% of transfected cells.

To assess light responsiveness (Supplementary Fig. 1c), we examined cluster disassembly after 15 min of light exposure. SNAP$_{tag}$-PixE/DsRed-PixD (1, 4) showed the most significant area decrease (1-$A_{15}/A_0 = 0.9401$) among all combinations including a multivalent protein; the mScarlet-PixE/mCerulean-PixE (1, 1) did not readily form clusters and thus it was excluded from this measurement (Supplementary Fig. 1c). Combinations involving FTL-PixD or FTL-PixE showed poor light responsiveness, possibly due to the strong multimerization effect of FTL; notably, FTL-PixE/mScarlet-PixD (24,1) showed an unexpected area increase (1-$A_{15}/A_0 = -1.657$) across all relevant combinations showed a decrease in area.

Evaluations of single-cluster size (Supplementary Fig. 1d) and the number of clusters per cell (Supplementary Fig. 1e) showed distinct patterns among the combinations. LanYFP-PixE/DsRed-PixD (4,4) and LanYFP-PixE/FTL-PixD (4,24) formed the largest clusters (1.687 and 1.587 μm², respectively) and the fewest clusters per cell (6.122 and 6.344, respectively). In contrast, FTL-PixE/DsRed-PixD (24,4) formed more and larger clusters than the other combinations (16.28 clusters/cell and 1.834 μm², respectively). Combinations including FTL-PixE tended to form larger clusters, prompting us to speculate that FTL-conjugated PixE may act as a seed to gather PixD.

Although FTL-PixE/DsRed-PixD (24,4) yielded many large clusters, their light responsiveness was low. We thus selected SNAP$_{tag}$-PixE/DsRed-PixD (1,4) as the basic framework for the RELISR system, due to its high cluster formation rate (85.7%) and excellent light responsiveness (95.57%).

Next, we aimed to verify that the selected (1,4) combination could be regulated under various light conditions. We expressed SNAP$_{tag}$-PixE and DsRed-PixD (1,4) in HeLa cells, confirmed that they successfully self-assembled in the dark state, and tested different protocols for

light stimulation. When we applied light stimulation of two pulses at intensities of 1, 2, 4, and 6 μW/cm², the results showed that stronger stimulation more effectively dissociated RELISR clusters, with the normalized cluster areas ($A_n/A_0$) of 0.86 at 1 μW/cm², 0.54 at 2 μW/cm², 0.40 at 4 μW/cm², and 0.36 at 6 μW/cm² (Supplementary Fig. 2a).

We further investigated the effect of varying the number of light pulses on cluster dissociation. When light pulses were applied repeatedly with intervals, we observed a progressive decrease in cluster area fraction with each subsequent pulse (Supplementary Fig. 2b). Additionally, applying consecutive pulses from 1 to 5 times led to a proportional decrease in cluster area, with more pulses causing a greater reduction (Supplementary Fig. 2c).

With whole-cell light illumination (488 nm; see "Methods"), clusters dissociated and diffused, resulting in an increase in cytosolic DsRed intensity (Fig. 1c, d). The cluster area decreased immediately upon light illumination (Fig. 1e), and cluster reassembly was detectable within 5 min after cessation of the light stimulation (Supplementary Movie 1).

To explore the spatiotemporal control capabilities of the RELISR system, we locally illuminated a single cluster in the subcellular region (Fig. 1f, Supplementary Movie 2). We expressed the RELISR system in HeLa cells and performed live-cell imaging of DsRed-PixD. In the dark, we identified the largest cluster in the cell as the "R1" region for local illumination, and designated a nearby "R2" region that contained two clearly visible clusters as the unstimulated control area (Fig. 1f). Upon local blue-light stimulation (see "Methods"), the R1 cluster completely disappeared within approximately 24 s of stimulation, whereas the two R2 clusters remained unchanged (Fig. 1f, h, i). We measured the DsRed fluorescence intensity in R1 and the cytosol outside of R1 (Fig. 1g). Under light stimulation, the DsRed signal in R1 rapidly decreased to nearly zero, while that in the cytosol increased 1.4-fold, indicating that DsRed-PixD dissociated and diffused into the cytoplasm rather than becoming photobleached.

## Protein storage and release using Protein-RELISR

To sequester proteins in the RELISR system, we selected as the cargo-binding domain the GFP nanobody, (V$_H$GFP)[19], which is a small antibody fragment capable of binding to GFP. We conjugated the GFP nanobody to the N-terminus of DsRed-PixD or SNAP$_{tag}$-PixE and assessed the GFP storage efficiency of each pair; V$_H$GFP-DsRed-PixD/SNAP$_{tag}$-PixE (V$_H$H-PixD) and V$_H$GFP-SNAP$_{tag}$-PixE/DsRed-PixD (V$_H$H-PixE) (Fig. 2a). We observed cluster formation in both groups and noted that these clusters colocalized with EGFP (Fig. 2b). The intensity profiles indicated that PixD and PixE colocalized well in both groups, with V$_H$H-PixD pairs showing lower EGFP signals in regions lacking clusters (Fig. 2c).

Consistently, Pearson correlation analysis[20] of the colocalization between EGFP and V$_H$H-PixD or V$_H$H-PixE (Fig. 2d) showed significantly higher colocalization of EGFP with V$_H$H-PixD compared to V$_H$H-PixE (Pearson coefficients $r$: 0.9574 and 0.6378, respectively, $P < 0.0001$). Additionally, V$_H$H-PixD formed significantly more clusters than V$_H$H-PixE (2.1-fold, $P = 0.0162$; Fig. 2e). There was no significant difference in cluster size between V$_H$H-PixD and V$_H$H-PixE ($P = 0.3132$; Fig. 2f). Considering the higher colocalization with the target GFP and the higher average cluster number, we selected V$_H$GFP-DsRed-PixD/SNAP$_{tag}$-PixE as the optimal configuration for the Protein-RELISR system.

We conducted time-lapse imaging to analyze the light responsiveness of the Protein-RELISR system. In the dark, EGFP, V$_H$GFP-DsRed-PixD and SNAP$_{tag}$-PixE colocalized and efficiently formed clusters. Upon light exposure, these clusters dissociated and the signals diffused into the cytosol within 5 min (Fig. 2g, i). The cytosolic fluorescence signals of DsRed and EGFP increased rapidly within 1 min of light exposure (maximum derivative values, DsRed: 172.99 RFU/min; EGFP: 67.89 RFU/min, Fig. 2h-Cytosol). Consistently, the cluster area

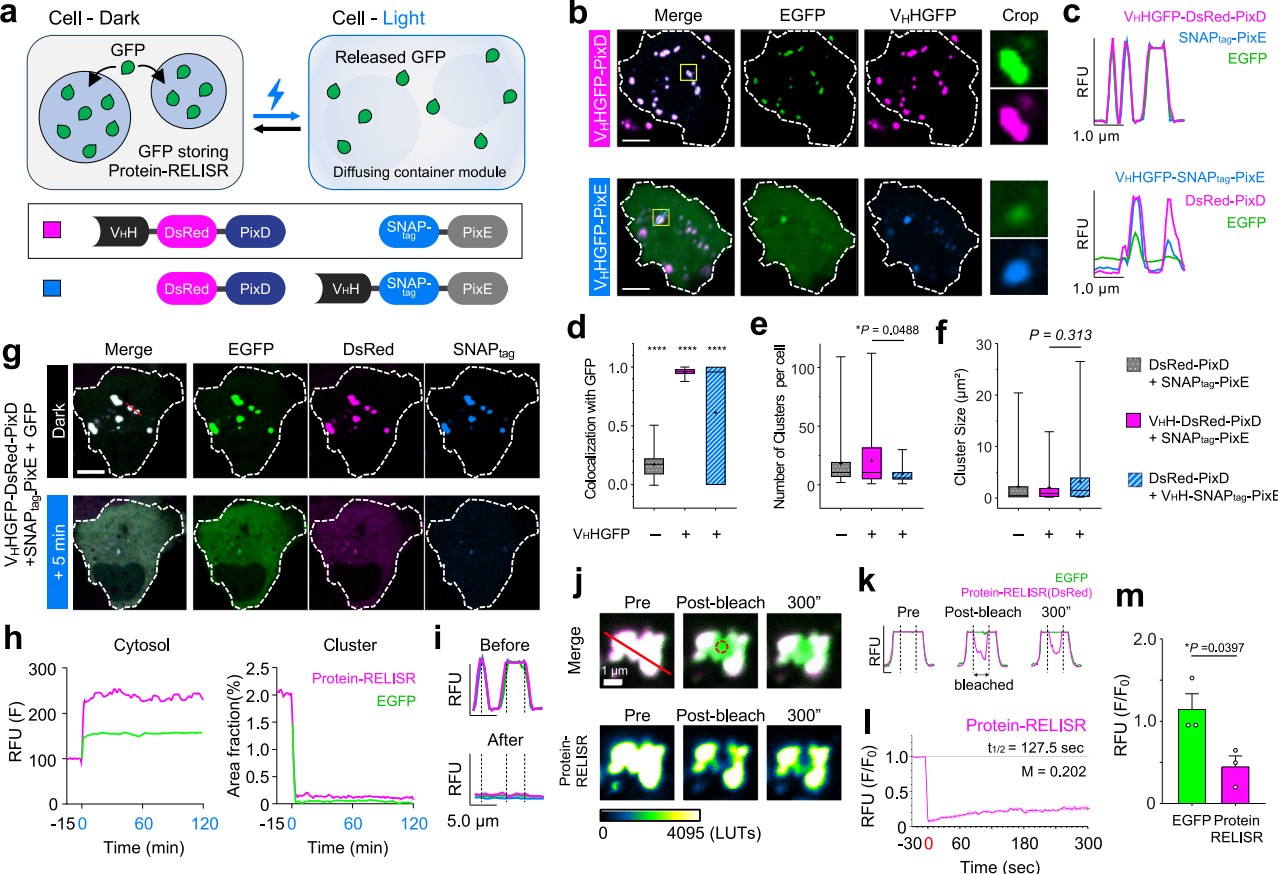

**Fig. 2 | Storage of GFP using protein-RELISR. a** Schematic of protein storage and release by Protein-RELISR, which dissociates upon blue light to release GFP-POI. Two construct pairs were tested: $V_H$HGFP-DsRed-PixD/SNAP$_{tag}$-PixE (magenta), and $V_H$HGFP-SNAP$_{tag}$-PixE/DsRed-PixD (blue). **b** Fluorescence images of HeLa cells expressing candidates and EGFP(Green). $V_H$HGFP-DsRed-PixD ($V_H$HGFP-PixD) / SNAP$_{tag}$-PixE (magenta) and $V_H$HGFP-SNAP$_{tag}$-PixE ($V_H$HGFP-PixE) / DsRed-PixD (blue). Yellow box: cropped region; white outline: cell boundary; blue line: profile axis (panel **c**). Scale bar, 10 μm. **c** Intensity profiles of panel (**b**): DsRed (magenta), SNAP$_{tag}$ (blue), and EGFP (green). (Top) $V_H$HGFP-PixD pairs; (Bottom) $V_H$HGFP-PixE pairs. **d**–**f**. Groups are indicated by colors and patterns: dark gray with dots (DsRed–PixD/SNAP$_{tag}$-PixE), solid magenta ($V_H$HGFP–PixD pairs), and hatched blue ($V_H$HGFP–PixE pairs). (one-way ANOVA, 50 clusters, 34 cells per group). (**d**) Mander's overlap coefficients. All groups differed significantly. (****$P$ < 0.00001). (**e**) Number of clusters per cell. **f** Single cluster size distribution. **g** Fluorescence images of HeLa cells expressing selected Protein-RELISR and target EGFP. The blue box shows the image at 5 min during light stimulation. White outline: cell boundary;

red line: profile axis (panel **i**). Scale bar, 10 μm. **h** Time-lapse graph of light-stimulation experiments of Protein-RELISR. DsRed (magenta) and EGFP (green) signals. From −15 min to 0 min shows the pre-stimulation session. Light stimulation started at 0 min and ended at 120 min. (Left) cytosolic intensity (sum excluding cluster regions). (Right) cluster area fraction over time (cluster area/cell area, %). **i** Intensity profiles before and after stimulation (from panel (**g**)). Magenta: $V_H$HGFP-DsRed-PixD; blue: SNAP$_{tag}$-PixE; green: EGFP. **j** Fluorescent images of FRAP experiments. "Pre" indicates before bleaching; "Bleach" shows immediately after; "300" marks 300 s post-bleaching. Merged images display DsRed (Protein-RELISR) and EGFP. DsRed is also shown as pseudo-colored single channels. Red circle: bleached region; red line: profile axis (panel **k**). **k** Intensity profile of the bleached cluster. **l** Time-lapse graph of FRAP experiments. DsRed (Protein-RELISR) fluorescence was normalized to the pre-bleach baseline. The graph begins 30 s before bleaching ("0" s). **m** Comparison for fold changes of fluorescence intensity (EGFP and Protein-RELISR) after photobleaching (unpaired two-tailed t-test, $n$ = 3 clusters; error bars, s.e.m.).

analysis showed that the DsRed and EGFP signals started to decrease sharply within 1.3 min (maximum negative derivatives, DsRed: −1.25%/min; EGFP: −1.29%/min, Fig. 2h-Cluster).

To assess whether Protein-RELISR forms stable and immobile condensates while sequestering EGFP, we performed the fluorescence recovery after photobleaching (FRAP) experiments[21] (Fig. 2j). We bleached $V_H$HGFP-DsRed-PixD within clusters and observed the changes in fluorescence intensities of both module and cargo (Fig. 2k, l). Upon bleaching, the normalized fluorescent intensity of $V_H$HGFP-DsRed-PixD immediately decreased ($F/F_0 = 0.244$), whereas that of EGFP remained unchanged ($F/F_0 = 1.144$), indicating that the cluster did not disassemble and release the cargo ($P = 0.0397$; Fig. 2m). At 10 min post-bleaching, the DsRed signal showed minimal recovery, with a maximum normalized intensity ($I_{max}$) of 0.283. Calculation of the mobile fraction of a cluster indicated that only 4.4% of single cluster was mobile (Fig. 2l). Similar to the FRAP results for the RELISR framework (Supplementary Fig. 3), these results indicate that there is

little exchange between $V_H$HGFP-DsRed-PixD in Protein-RELISR. Thus, RELISR exhibits restricted internal dynamics that enable efficient cargo protein sequestration.

To test whether Protein-RELISR can repeatedly store and release GFP, we performed a three-sets of light stimulation period (10-min) and dark period (20-min; Supplementary Fig. 4). Upon each light stimulation, the cytosolic DsRed signal of Protein-RELISR increased ($F_{10}/F_0 = 1.307$, $F_{40}/F_0 = 1.232$, $F_{70}/F_0 = 1.297$), and subsequently decreased during the dark periods ($F_{30}/F_0 = 1.081$, $F_{60}/F_0 = 1.159$, $F_{90}/F_0 = 1.121$), indicating reversible condensate disassembly and reassembly. Similarly, cytosolic GFP intensity rose during illumination (GFP: $F_{10}/F_0 = 1.164$, $F_{40}/F_0 = 1.167$, $F_{70}/F_0 = 1.219$) and declined during the following dark intervals ($F_{30}/F_0 = 1.072$, $F_{60}/F_0 = 1.134$), demonstrating functional reversibility even after repeated stimulation (Supplementary Fig. 4b).

We next wondered whether the RELISR system can store and release multimeric cargoes. We compared three GFP nanobody

conjugated constructs: $V_H$HGFP-DsRed-PixD ($V_H$H-PixD), $V_H$HGFP-SNAP$_{tag}$-PixE ($V_H$H-PixE), and a tandem GFP nanobody conjugated, $V_H$HGFP-$V_H$HGFP-SNAP$_{tag}$-PixE ($V_H$Hx2-PixE). As a multimeric target, we conjugated EGFP to the association domain of CaMKIIα (AD), which forms a dodecamer[22]. We targeted both monomeric EGFP and dodecameric EGFP-AD as cargo (Supplementary Fig. 5a, b).

To assess EGFP sequestration, we measured the colocalization between EGFP and each construct using Pearson's correlation coefficient ($r$). $V_H$H–PixD showed the highest colocalization ($r = 0.8397$), significantly higher than both $V_H$H–PixE ($r = 0.7309$; $P = 0.0375$) and $V_H$Hx2–PixE ($r = 0.6918$; $P = 0.0013$). Colocalization differences between $V_H$H–PixE and $V_H$Hx2–PixE were not statistically significant ($P = 0.5386$; Supplementary Fig. 5c). These data suggest that $V_H$H–PixD provides the most efficient sequestration of monomeric EGFP cargo.

Next, we examined EGFP release dynamics under moderate light stimulation (see Methods; Supplementary Fig. 5d). Cluster areas were normalized, and dissociation kinetics were analyzed using one-phase decay models. The rate constant $K$, which reflects the speed of cargo release (with higher values indicating faster dissociation), was highest for $V_H$H–PixD ($K = 10.52$), followed by $V_H$Hx2–PixE ($K = 5.498$) and $V_H$H–PixE ($K = 2.031$), indicating its efficient EGFP sequestration and prompt release.

In contrast, when targeting multimeric EGFP-AD (Supplementary Fig. 5b), $V_H$Hx2-PixE achieved the highest colocalization ($r = 0.9060$), followed by $V_H$H-PIXD ($r = 0.8797$) and $V_H$H-PIXE ($r = 0.8508$), although the differences were not statistically significant (n.s., all $P > 0.2$; Supplementary Fig. 5c). Release kinetics revealed significant differences among constructs: $V_H$H–PixD followed a one-phase decay model, whereas both $V_H$Hx2–PixE and $V_H$H–PixE followed a two-phase decay model (Supplementary Fig. 5e). $V_H$Hx2-PixE showed the fastest dissociation ($K_1 = 348.1$), while $V_H$H-PixE displayed moderate kinetics ($K_1 = 81.37$). In contrast, $V_H$H-PixD dissociated slowly ($K = 7.075$), retaining ~89% of aggregates after extended illumination (Supplementary Fig. 5e). These findings suggest that while $V_H$H-PixD is optimal for monomeric cargo, $V_H$Hx2-PixE or $V_H$H-PixE may be more suitable for highly multimeric targets. Notably, $V_H$HX2-PixE did not outperform $V_H$H-PixD in the monomeric context, indicating that tandem nanobody design does not universally improve performance. Together, these results underscore the need to match construct architecture with cargo to ensure effective sequestration and controlled release.

## Protein storage and release in neurons

To expand the applicability of our system, we investigated whether Protein-RELISR could function within complex structures, such as dense or narrow spaces. As a model, we selected a rat hippocampal neuron. Neuronal cells have a well-compartmentalized structure that can be divided into the soma and long, narrow neurites, wherein biomolecules must be efficiently packed and transported to the termini for neuronal survival and functions[23] such as synaptic transmission and plasticity[24]. To precisely modulate biomolecules in neurites, proteins must be sequestered and released in a controlled manner. Therefore, we introduced the Protein-RELISR system into the neuron to explore its ability to enable finely tuned protein storage and release within this complex architecture (Fig. 3a).

We transfected neurons with RELISR (Supplementary Fig. 6a) or Protein-RELISR (Supplementary Fig.6b), both with EGFP. Clusters were observed throughout the neurons of both groups; signals were seen in the soma and neurites, with larger clusters in the soma (Fig. 3b and Supplementary Fig. 6c). Protein-RELISR clusters in both the soma and neurites colocalized with EGFP (soma: $r = 0.8585$, neurites: $r = 0.9095$; Supplementary Fig. 6d). There was no significant difference in targeting efficiency by cluster location ($P = 0.3489$). RELISR without the GFP nanobody showed significantly less GFP colocalization than Protein-RELISR ($P = 0.0044$; Fig. 3c), as seen in the whole cell, soma, and neurites (Fig. 3c, and Supplementary Fig. 6d).

To show the spatial resolution of RELISR in the complex structure of neurons, we conducted local stimulation in the soma or neurites of a given cell (Fig. 3d, Supplementary Movie 3). Local illumination to the region of interest (Fig. 3e, blue circle, stimulated ROI) dissolved the DsRed and EGFP clusters (Fig. 3g and i) and GFP signal decreased (GFP: $F_{25}/F_0 = 0.174$; Fig. 3k). In contrast, GFP fluorescence temporarily increased in the adjacent ROI (GFP: $F_{25}/F_0 = 1.407$). This indicates that Protein-RELISR successfully released EGFP locally in the soma upon light stimulation. The same results were obtained from parallel experiments performed in the corresponding neurites (Fig. 3f, h, j, l). These findings confirm that Protein-RELISR can effectively store protein even in a complex structure like neuron, and release them successfully upon local light stimulation.

## Protein-RELISR enables light-induced functional protein activation

To investigate whether Protein-RELISR could modulate cellular processes via the light-induced release of functional proteins, we used the protein Vav2[25], a guanine nucleotide exchange factor that interacts with Rac1 and induces cell protrusion. We hypothesized that the storage and release of Vav2 by Protein-RELISR would enable us to direct light-dependent changes in cell morphology (Fig. 4a). We tagged EGFP to the N termini of Vav2 with a membrane localization motif, Rit-tail[26] at C-termini (EGFP-Vav2).

We expressed RELISR or Protein-RELISR with the cargo EGFP-Vav2 in NIH3T3 mouse fibroblast cells (Fig. 4b, c). Consistent with the above-described results (Fig. 2d), significant signal colocalization was observed when GFP nanobody was present (Fig. 4d). FRAP experiments with the Vav2-targeting Protein-RELISR confirmed that only 1.6% of a cluster was mobile, indicating that the cluster were stably formed and trapped Vav2 (Supplementary Fig. 7).

To confirm that Protein-RELISR effectively stored the cargo in the dark state, we assessed cell circularity[27], which measures how closely the shape of a marked region resembles a circle, with values ranging from 0 to 1 (Fig. 4e). Overexpression of Vav2 is known to result in a rounded cell shape[25,28]; therefore, any leaky storage of Vav2 would lead to alteration in cell morphology. We examined cells expressing Protein-RELISR/EGFP, Protein-RELISR/EGFP-Vav2, and RELISR/EGFP-Vav2 (without GFP nanobody) (Fig. 4e). There was no significant difference in circularity between cells expressing Protein-RELISR/EGFP and Protein-RELISR/EGFP-Vav2 (0.4131 and 0.3365, respectively, $P = 0.5039$), indicating that Protein-RELISR successfully stored Vav2 in the dark. In contrast, RELISR/EGFP-Vav2 expressing cells exhibited significantly higher circularity (0.5648, Fig. 4b, e, P < 0.0001), indicating that Vav2 was not effectively stored without the GFP nanobody. These results confirm that Protein-RELISR stores with the cargo protein and restricts its function via sequestration.

We next performed light stimulation of cell expressing Protein-RELISR/EGFP-Vav2 and assessed the changes in circularity during the periods. Upon illumination, clusters disassembled, and EGFP-Vav2 diffused throughout the cell. This caused overall cell protrusion (Fig. 4c) and a significant 1.3-fold increase in cell circularity ($P = 0.0459$, Fig. 4f). When the light was turned off, small clusters reassembled at the protrusion sites and the protruded membrane collapsed (Supplementary Movie 4). An hour after the light was turned off, the circularity decreased to levels similar to those before illumination ($P = 0.5125$; Fig. 4f). These observations confirm that the Protein-RELISR system can effectively store and re-store EGFP-Vav2 in the dark state and release the cargo in a functional form upon light stimulation.

To demonstrate the versatility of Protein-RELISR, we employed various GFP-tagged Rho GTPases and GEFs, including Tiam1, Rac1, Cdc42, and RhoG (Fig. 4g)[25,28-33]. We expressed each GFP-conjugated cargo along with RELISR or Protein-RELISR. All GFP-conjugated proteins strongly colocalized with Protein-RELISR, and their sequestration

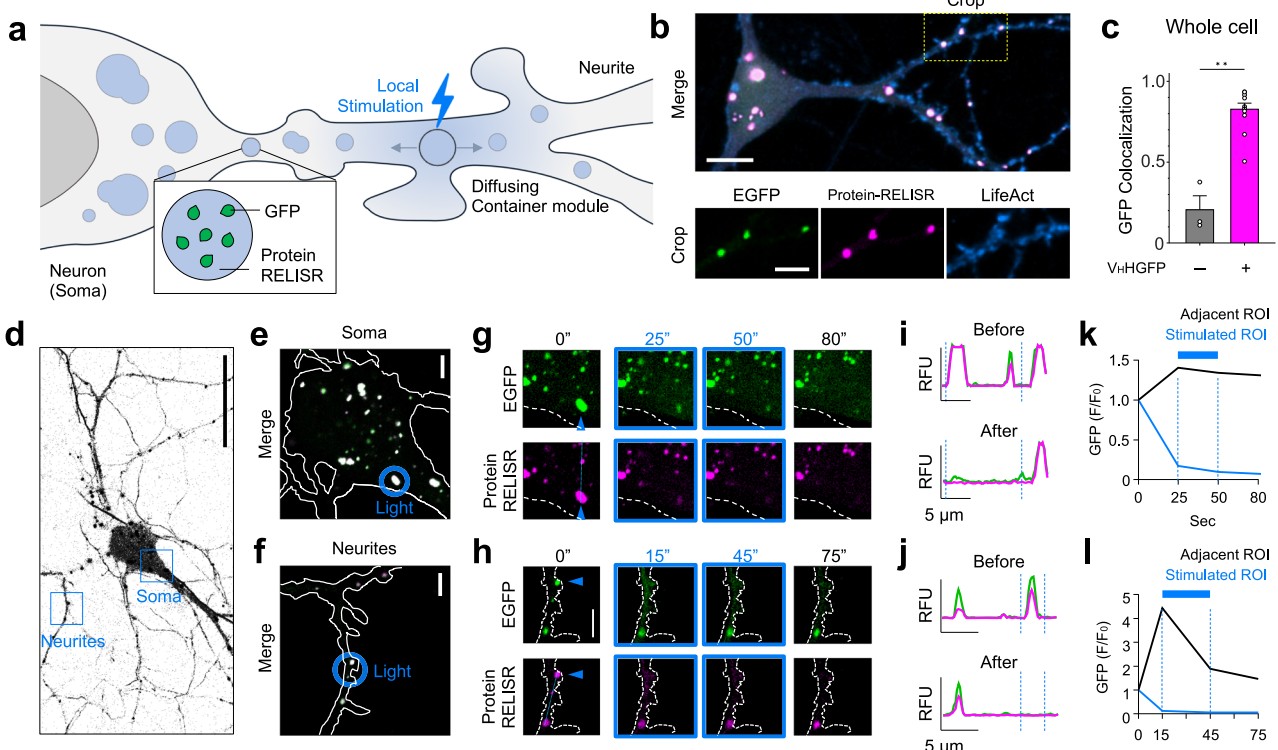

**Fig. 3 | RELISR-mediated protein release in complex structures of neurons.**
**a** Illustration of how Protein-RELISR works in complex structures within a neuron. The GFP-storing container dissociates with local light stimulation. **b** Fluorescence image of a neuron expressing Protein-RELISR. (Top) Merged channel image. Yellow dotted box denotes the cropped area. (Bottom) The cropped image. EGFP (green), Protein-RELISR (DsRed, magenta), and LifeAct (iRFP682, blue). Scale bar, 5 μm. **c** Pearson correlation analysis of GFP colocalization with RELISR (gray) or Protein-RELISR (magenta), calculated across entire cells. ($n = 3$ cells for RELISR; $n = 12$ cells for Protein-RELISR, each from independent experiments; **$P = 0.0044$, Mann-Whitney test, error bars, s.e.m.). **d** Fluorescence images of neurons in inverted mono. Blue boxes show the cropped area for "Soma" and "Neurites" in panels (**e**) and (**f**). This is a representative image from three independent experiments with similar results. Scale bar, 50 μm **e, f** Cropped images from panel (**d**) Merged channel images of soma (**e**) and neurites (**f**). White dotted lines show the outline of

the cell. Blue circles indicate the local light-stimulated region (Stimulated ROI). Scale bar, 5 μm. **g, h** Time-lapsed images, cropped from soma (**g** from (**e**) and neurites (**h**) from (**f**) during local stimulation with 0.3 laser intensity. "0 s" shows right before local light stimulation. Blue triangles indicate stimulated clusters. Blue dotted line indicates where intensity profiles for panels (**i**) and (**j**) were analyzed. Blue boxed images (25 s, 50 s for soma and 15 s, 45 s for neurites) are from the stimulation sessions. 80 s and 75 s show images after local light stimulation. Scale bar, 5 μm. **i, j** Intensity profiles of (**g**) and (**h**). The areas between the two blue dotted lines show the stimulated region in (**g**) and (**h**). Protein-RELISR (magenta) and EGFP (green). Scale bar, 5 μm. **k, l** GFP intensity fold changes over time in soma (**k**) and neurites (**l**). The fluorescent intensity of the adjacent ROI (black) was calculated by subtracting the sum of fluorescent intensity in the stimulated ROI from that of the whole cell area. Stimulated ROI for blue.

required the GFP nanobody (Supplementary Fig. 8). We observed a significant increase in the area of cells only in the Protein-RELISR group upon light illumination (Tiam1, $P < 0.0001$; Cdc42, $P < 0.0015$; Rac1, $P < 0.0028$; Fig. 4g). These results show that Protein-RELISR can store and release various GFP-conjugated proteins in response to light, leading to functional changes in entire cells.

Next, we investigated whether Protein-RELISR enables local regulation of signaling proteins in neurons, adopting GFP conjugated PTEN[34,35], a phosphoinositide phosphatase that dephosphorylates PIP$_3$ to PIP$_2$ (Supplementary Fig. 9a). PTEN-GFP, V$_H$HGFP–DsRed–PixD, SNAP$_{tag}$–PixE, and iRFP682–PLCδ-PH domain were co-expressed in primary hippocampal neurons to monitor PIP$_2$ dynamics in response to targeted blue-light stimulation (Supplementary Fig. 9b). Local illumination of neurites triggered the release of PTEN-GFP from RELISR clusters, resulting in decreased cytosolic PLCδ−PH intensity (cytosolic $I_{10}/I_0 = 0.8314$) and increased membrane-associated signal ($I_{10}/I_0 = 1.3543$), with a significant difference between regions ($P = 0.0205$; Supplementary Fig. 9f). The cytosolic PLCδ−PH signal in the stimulated region also decreased significantly compared to the non-stimulated region ($P < 0.0001$; Supplementary Fig. 9d), consistent with localized PTEN activity that reduces PI(3,4,5)P$_3$ levels and increases PI(4,5)P$_2$ at the site of release. These observations demonstrate that Protein-

RELISR facilitates spatiotemporal release and activation of functional protein cargo in neurons.

## Light-induced target translation using mRNA-RELISR
mRNA translation is precisely controlled to ensure rapid and efficient protein production at specific locations and times[36–38]. Cells sequester mRNA within specialized compartments called ribonucleoprotein (RNP) granules to prevent interactions with the translational machinery and thereby regulate translation spatially and temporally[5,39,40]. Inspired by these RNP granules, we aimed to store mRNA in RELISR clusters and release it upon light stimulation to trigger translation of the cargo mRNA (Fig. 5a). As a cargo binding domain, we employed the MS2-coated protein (MCP)[41,42], which selectively interacts with the secondary stem-loop structure known as the MCP-binding site (MBS) in mRNA. We conjugated tandem MCP (tdMCP)[43] to the N-terminus of DsRed-PixD and targeted CFP-MBSx24[43], as the cargo mRNA with 24 MBS inserted into the 3′UTR of the sequence encoding CFP (Fig. 5b).

To investigate the effect of tdMCP and the cargo mRNA on the cluster formation of RELISR, we co-expressed RELISR or tdMCP-conjugated RELISR (mRNA-RELISR) with CFP lacking MBS or CFP-MBS in HeLa cells and performed confocal microscopic imaging. Clusters formed well in all groups (Fig. 5c, d). Interestingly, the presence of

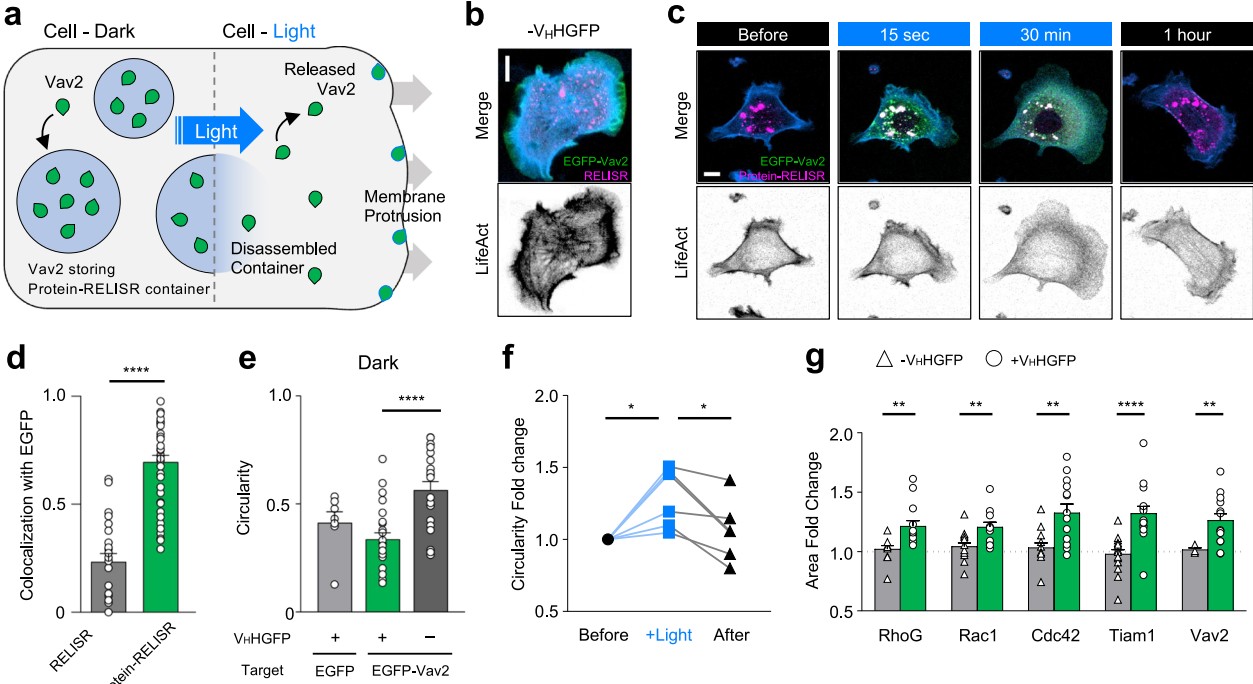

**Fig. 4 | Light-induced release of functional protein in fibroblast cells.**
**a** Schematic illustration of Protein-RELISR in a fibroblast cell. (Left) In the dark, the target protein (Vav2) is stored and inhibited. (Right) With light illumination, Vav2 is released and activated, inducing membrane protrusion. **b** Fluorescence images of NIH3T3 cells expressing EGFP-Vav2 and the RELISR system without the GFP nanobody. Inverted monochrome channel image shows iRFP682-LifeAct. Scale bar, 10 μm This is a representative image from at least three independent experiments with similar results. **c** Time-lapse images of NIH3T3 cells expressing EGFP-Vav2 and the Protein-RELISR system. "Before" indicates before light stimulation. "15 s" and "30 min" indicates the time point during light stimulation. "After" indicates ab hours of recovery in the dark. Colored images include merged channels: EGFP-Vav2 (green), Protein-RELISR (DsRed, magenta), and iRFP682-LifeAct (blue). Inverted monochrome channel images show iRFP682-LifeAct. Scale bar, 10 μm **d** Colocalization analysis of two groups: RELISR (-V$_H$HGFP) and Protein-RELISR ( + V$_H$HGFP). Pearson's correlation coefficients ($n = 24$ cells for each group, ****$P < 0.00001$, unpaired two-tailed t-test, error bars, s.e.m.). **e** Circularity evaluation in the dark. Gray for Protein-RELISR with EGFP, green for Protein-RELISR with EGFP-Vav2, and dark gray for RELISR with EGFP-Vav2 ($n = 7, 25, 21$ cells for each group, ****$P = 0.000038$, one-way ANOVA, error bars, s.e.m.). **f** Quantifications of circularity fold change during light stimulation experiments. Each circularity was normalized by initial value. "Before" for black circle, "+Light" for blue square, and "After" for black triangle. Each cell is connected by a line with each symbol ($n = 6$ cells, *$P = 0.0459$, one-way ANOVA). **g** Comparison of the cell area fold changes after light stimulation between RELISR (triangle) and Protein-RELISR (circle). RhoG (**$P = 0.0029$, $n = 13$ cells for RELISR and 16 cells for Protein-RELISR), Rac1 (**$P = 0.0028$, $n = 15, 14$), Cdc42 (**$P = 0.0014$, $n = 14, 14$), Tiam1 (****$P = 0.0000274$, $n = 16, 16$), Vav2 (**$P = 0.0094$, $n = 6, 14$). unpaired two-tailed t-test, error bars, s.e.m.

tdMCP increased the number of clusters (RELISR/CFP: 9.722, mRNA-RELISR/CFP: 20.14), possibly via MCP-driven dimerization (Fig. 5d). Additionally, in the presence of cargo mRNA (CFP-MBS), mRNA-RELISR formed significantly more clusters (mRNA-RELISR/CFP: 20.14, mRNA-RELISR/CFP-MBS: 30.83, $P = 0.0034$, Fig. 5c, d). This phenomenon likely reflects that the 24 MBS enable numerous multivalent interactions[44] with tdMCP-DsRed-PixD, thereby increasing the cluster formation events at multiple sites, and resulting in a higher number of clusters.

We analyzed the fluorescent signal intensity of CFP, which was encoded by the cargo mRNA (Fig. 5e). In the RELISR group (without tdMCP), there was no significant difference in CFP signal with or without MBS ($P = 0.1840$). However, in the mRNA-RELISR group with tdMCP, the CFP signal intensity was significantly lower in the presence of MBS ($P = 0.0002$, Fig. 5c, e), indicating that CFP-MBS mRNA translation was inhibited and thus the cargo was sequestered.

To evaluate the light responsiveness of mRNA-RELISR, we co-expressed mRNA-RELISR and the cargo mRNA (CFP-MBS) in HeLa cells. We monitored DsRed signal intensity from 1 h before light exposure, exposed them to light for 2 h, and continued monitoring for 2 h after light was turned off (total, 5 h; Fig. 5f,g, Supplementary Movie 5). Upon light exposure, the intensity of mRNA-RELISR clusters decreased to nearly half the initial level within 25 min (Cluster $F_{23}/F_{-60} = 0.508$). As the clusters disassembled and diffused into the cytosol, the cytosolic DsRed intensity increased 2.56-fold (Cytosolic $F_{120}/F_{-60}$). After light was turned off, the cluster reassembled and the intensity gradually

increased to nearly the initial level (Cluster $F_{240}/F_{-60} = 0.989$). Therefore, the cytosolic intensity, which had risen during illumination, decreased from 2.56- to 1.74-fold (Cytosolic $F_{240}/F_{-60}$). These results confirmed that mRNA-RELISR clusters exhibited light-dependent disassembly and reassembly.

We further analyze whether mRNA-RELISR has different kinetics from that of Protein-RELISR (Supplementary Fig. 11). A comparison between Protein-RELISR and mRNA-RELISR reveals that mRNA-RELISR exhibited a higher dissociation rate ($K = 16.26$) than Protein-RELISR ($K = 10.52$; Supplementary Fig 11a). However, mRNA-RELISR reached plateau ($A_1/A_0 = 0.4960$) in 40 min (when the logarithmic rate of decay dropped below $10^{-4}$; Supplementary Fig. 11a), whereas Protein-RELISR targeting GFP did not reach the plateau within our time window (see Methods). Repeated light stimulation confirmed that mRNA-RELISR consistently responded to light stimulation ($F_{10}/F_0 = 2.791$, $F_{30}/F_0 = 2.026$, $F_{40}/F_0 = 2.993$, $F_{60}/F_0 = 2.290$, $F_{90}/F_0 = 3.284$, $F_{110}/F_0 = 2.284$), demonstrating its reversibility and light-responsiveness (Supplementary Fig. 11b). Further comparison of mobile fractions during FRAP experiments revealed a significant difference between Protein-RELISR and mRNA-RELISR (Fig. 2l, Supplementary Fig. 10, 11c), with Protein-RELISR showing a higher mobile fraction than mRNA-RELISR (Protein-RELISR: 0.2027, mRNA-RELISR: 0.0577, $P = 0.0189$; Supplementary Fig. 11c). These results characterize the light-responsiveness behavior and cluster stability of mRNA-RELISR, although it differs from Protein-RELISR.

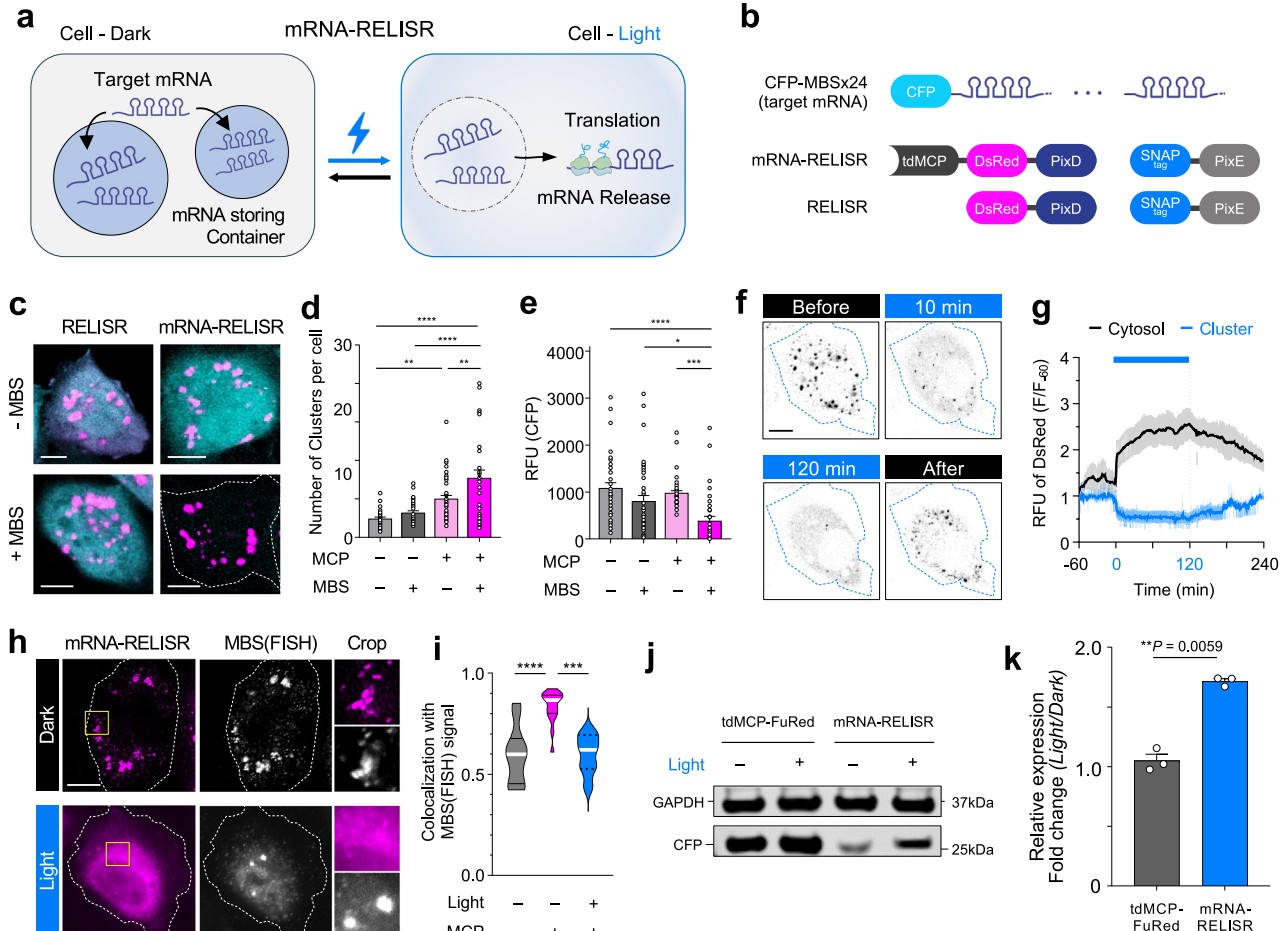

**Fig. 5 | Target mRNA storage and release by mRNA-RELISR. a** Schematic representation of target mRNA storage and release (mRNA-RELISR). Stored mRNA is released upon light illumination, enabling its translation. **b** Constructs of target mRNA, mRNA-RELISR, and RELISR. CFP transcripts carry 24 x MBS in the 3′UTR. tdMCP is fused to the N-terminus of DsRed-PixD in mRNA-RELISR. **c** Merged channel images of HeLa cells expressing RELISR or mRNA-RELISR. DsRed: magenta; CFP: cyan. "−MBS" indicates CFP mRNA without MBS; "+MBS," CFP mRNA with 24 × MBS. White outline: cell boundary. Scale bar, 10 μm **d, e.**= Quantification of experiments in panel (**c**). RELISR + CFP ($n = 36$ cells/group, gray), RELISR + CFP-MBS ($n = 36$, dark gray), mRNA-RELISR + CFP ($n = 42$, pale pink), and mRNA-RELISR + CFP-MBS ($n = 30$, pink). One-way ANOVA, error bars, s.e.m. (**d**). Number of clusters per cell (**P = 0.0025, **P = 0.0034, **** P < 0.00001). (**e**) CFP intensity (***P = 0.0002, *P = 0.0166, ****P < 0.00001). **f** Time-lapse images of an mRNA-RELISR-expressing HeLa cell under light stimulation. DsRed signals shown. "Before" indicates pre-stimulation; "10 min" and "120 min" indicate 10 min and 2 h of stimulation; "After" shows recovery, 2 h after stimulation in the dark. Scale bar, 10 μm

**g** Time-lapse graph of panel f. RFU (F/F$_{-60}$) indicates normalized DsRed intensity. Bold blue line marks the light-stimulation period. Black: cytosol; blue: clusters. Shaded areas (gray, pale blue; error bands) show mean ± s.e.m. **h** Fluorescence in situ hybridization (FISH) images of mRNA-RELISR (magenta) and CFP-MBS mRNA (white, Quasar 670 probe). Yellow boxes: cropped areas; white outlines: cell boundaries. Scale bar, 10 μm. **i**. Violin plot of colocalization between mRNA-RELISR and MBS FISH signals. Gray: RELISR (dark); pink: mRNA-RELISR (dark); blue: mRNA-RELISR (light). White bold line: median. (****P = 0.00001, ***P = 0.0001, n = 12, 15, 7 cells/group, one-way ANOVA). **j**. Immunoblot of CFP expression under light stimulation (see Methods). GAPDH (37 kDa), CFP (27 kDa). "tdMCP-FuRe": CFP-MBSx24 + tdMCP-FusionRed; "mRNA-RELISR": CFP-MBSx24 + mRNA-RELISR. **k** Quantification of relative CFP expression from immunoblot. Intensities of CFP were normalized to that of GAPDH, and fold change was calculated as (CFP/GAPDH)$_{Light}$ / (CFP/GAPDH)$_{Dark}$. ($n = 3$ biological replicates/group, paired two-tailed t-test, error bars, s.e.m.).

To assess the sequestration of CFP-MBS within mRNA-RELISR, we performed Fluorescence In Situ Hybridization (FISH)[45] using probes targeting the three different sequences of MBS[46]. We observed the localization of mRNA-RELISR and CFP-MBS under both dark and light conditions (Fig. 5h). In the dark group, the DsRed signal of mRNA-RELISR and the Alexa647 probe-stained MBS FISH signal showed much higher colocalization than the RELISR group without tdMCP (mRNA-RELISR: $r = 0.8332$, RELISR: $r = 0.5936$, respectively, Fig. 5i). In the light-stimulated group, the mRNA-RELISR cluster dissociated into smaller spots that either remained near MBS FISH signals or diffused throughout the cytosol, leading to significant decrease in the colocalization coefficient with MBS FISH signals ($r = 0.5976$, P = 0.0001; Fig. 5i and Supplementary Fig. 12a). To verify that mRNA-RELISR specifically targeted MBS in the dark, we conducted FISH for the

endogenous housekeeping gene, GAPDH, which lacks an MBS (Supplementary Fig. 12b). mRNA-RELISR did not colocalize with GAPDH in the dark ($r = 0.4013$, Supplementary Fig. 12c). These results confirmed that mRNA-RELISR specifically targets MBS-containing mRNA and releases them upon light exposure.

Finally, we employed immunoblot analysis to determine whether mRNA-RELISR could specifically induce the translation of target mRNA upon light exposure (Fig. 5j). Based on previous studies showing that tdMCP affects mRNA stability upon MBS binding[41,47], we transfected CFP-MBS and tdMCP-FusionRed instead of mRNA-RELISR for the control group. The control groups showed no significant difference in CFP levels between dark condition and light-stimulated condition (Fold change = 1.050, Fig. 5j, k). As expected, mRNA-RELISR showed a significant increase in CFP translation upon light stimulation (Fold change = 1.716, P = 0.0059, Fig. 5j, k). This demonstrates that mRNA-

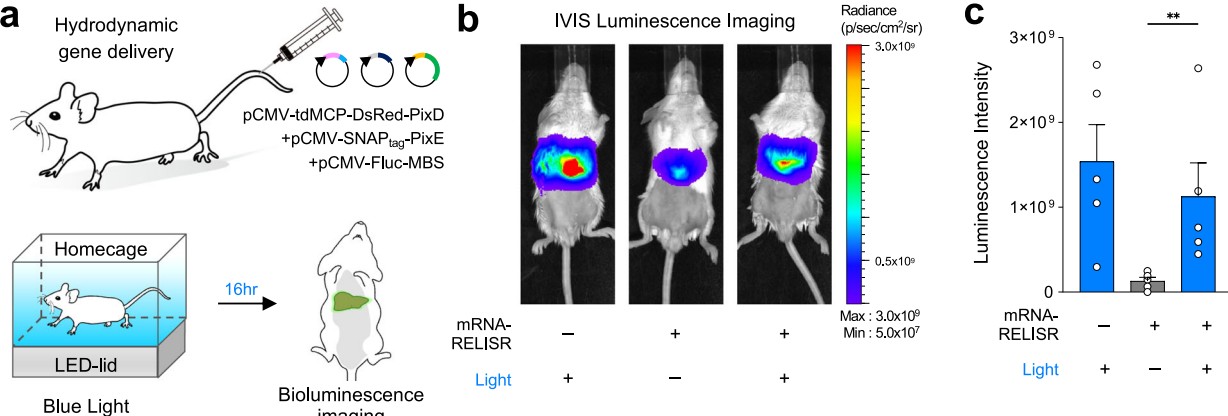

**Fig. 6 | Light-induced target mRNA translation in vivo. a** Schematic overview of the in vivo experiments. mRNA-RELISR and MBS-tagged Firefly Luciferase (Fluc-MBS) were transfected via hydrodynamic gene delivery (tail-vein injection), and belly-shaved mice were illuminated using an LED-lid under their home-cage. Bioluminescence imaging was carried out 16 hrs after the injection. Luciferin was injected right before imaging. **b** Luminescence images of mice with mRNA-RELISR only (Dark), mRNA-RELISR with Fluc-MBS (Dark), and mRNA-RELISR with Fluc-MBS (Light). Color scale bar proceeds from violet ($5.0 \times 10^7$) to red ($3.0 \times 10^9$). **c** Luminescence intensity bar graph. The mRNA-RELISR-only group did not show any luminescence. Gray indicates the dark group and blue indicates the light-stimulated group ($n = 5$ mices/group, **$P = 0.0079$, two-tailed Mann-Whitney test, error bars, s.e.m.).

RELISR controlled the cargo mRNA (CFP-MBS) translation by storing and releasing it in a light-dependent manner.

We further assessed whether the translational regulation by mRNA-RELISR condensates exceeds that achieved by altering mRNA localization alone. For comparison, we used mitochondrial LOVTRAP (mito-LOVTRAP)[48,49], which anchors Zdk1-fusion proteins to mitochondria in the dark, spatially restricting their function. Blue-light illumination dissociates LOV2–Zdk1 complexes, releasing cargo proteins into the cytoplasm. We hypothesized that tdMCP-conjugated LOVTRAP would alter the localization of MBS-tagged mRNA in a light-dependent manner (Supplementary Fig. 13). We co-expressed Tom20–mVenus–LOV2 (Tom20–LOV2) with either tdMCP–FusionRed–Zdk1 (tdMCP–Zdk1) or mScarlet–Zdk1–tdMCP (Zdk1–tdMCP) in HeLa cells (Supplementary Fig. 13a, b). In the dark state, both tdMCP–Zdk1 and Zdk1–tdMCP colocalized with Tom20–LOV2 at mitochondria. Upon blue-light illumination (457 nm; see Methods), both groups dispersed into the cytoplasm and subsequently relocalized to mitochondria upon returning to darkness, confirming light-dependent localization independent of fusion orientation (Supplementary Fig. 13a, b).

Next, we evaluated whether mito-LOVTRAP systems effectively control mRNA localization using fluorescence in situ hybridization (FISH) of CFP–MBS transcripts (Supplementary Fig. 13c–e). Under dark conditions, CFP–MBS transcripts showed moderate colocalization with Tom20–LOV2 (tdMCP–Zdk1: $r = 0.6482$; Zdk1–tdMCP: $r = 0.6222$; n.s., P = 0.9982). Light exposure slightly reduced colocalization (tdMCP–Zdk1: $r = 0.5248$; Zdk1–tdMCP: $r = 0.5175$), though these differences were not statistically significant (tdMCP–Zdk1: n.s., P = 0.0767; Zdk1–tdMCP: n.s., P = 0.0960; Supplementary Fig. 13e).

We then assessed whether mito-LOVTRAP-based constructs could regulate mRNA translation (Supplementary Fig. 13f, g). Western blot analysis revealed minimal differences in CFP protein levels between dark and illuminated conditions (tdMCP–FusionRed control: 1.0528; tdMCP–Zdk1: fold change = 0.9809; Zdk1–tdMCP: 1.020; all $P > 0.79$; Supplementary Fig. 13g). Thus, mito-LOVTRAP systems fused to tdMCP do not significantly regulate cargo mRNA translation.

These findings indicate that while mito-LOVTRAP effectively controls protein localization[48,49], it does not physically isolate mRNA from the translational machinery. In contrast, RELISR condensates physically sequester mRNA, enabling robust translational suppression in the dark and efficient reactivation upon illumination (Fig. 5j, k

and Fig. 6). Consistent with this mechanistic distinction, our direct comparison showed mito-LOVTRAP exerted no significant effect on translation under dark conditions (Supplementary Fig. 13f, g). Thus, mRNA-RELISR offers a distinct sequestration strategy—physical exclusion via condensate formation—crucial for effective translation control. These results suggest that although mito-LOVTRAP effectively controls protein localization, it does not physically isolate mRNA from the translational machinery. In contrast, RELISR condensates physically sequester mRNA, enabling robust translational suppression in darkness and efficient reactivation upon illumination (Fig. 5j, k). Consistent with this mechanistic distinction, our direct comparison showed mito-LOVTRAP had no significant effect on translation under dark conditions (Supplementary Fig. 13f, g). Thus, RELISR provides a distinct sequestration strategy—physical exclusion through condensate formation—which is crucial for effective translation control.

## Light-induced translation by mRNA-RELISR in an in vivo mouse model

Next, we set out to demonstrate that our system functions in vivo, and thus may hold potential to be developed for therapeutic applications. A previous study demonstrated that an optogenetic tool can be successfully expressed and activated in living mice using blue light[50]. For the present work, we selected Firefly luciferase[51] (Fluc) as the cargo mRNA, and attached 24 MBS to its 3' UTR.

To express the mRNA-RELISR system in a mouse model, we utilized the hydrodynamic gene delivery system. We delivered mRNA-RELISR and Fluc-MBS via tail-vein injection into 6-week-old male BALB/c mice (Fig. 6a), which were distributed to three groups: Fluc-MBS(Cargo) only, mRNA-RELISR/Fluc-MBS without light exposure, and mRNA-RELISR/Fluc-MBS with light exposure ($n = 5$ mice/group). At 16 h post-injection, the mice were anesthetized and abdominally shaved, and luminescence imaging was performed. mRNA-RELISR/Fluc-MBS mice with light exposure exhibited ~8.5 times higher luminescence than mRNA-RELISR/Fluc-MBS mice without light exposure ($P = 0.038$, Fig. 6b, c). This indicates that mRNA-RELISR effectively stores cargo mRNA in the dark and releases it upon light stimulation, leading to efficient translation of target mRNA in a mouse model. Importantly, we demonstrate that artificial condensates can be engineered to release mRNA capable of inducing translation in vivo.

## Discussion

We developed RELISR, an optogenetic condensate platform constructed by fusing an oligomeric protein and cargo-binding domains to a light-sensitive switch, enabling cargo sequestration in the dark and release upon blue-light illumination. RELISR provides precise spatiotemporal control of protein function and mRNA translation.

We optimized the basic framework by varying the valency of the proteins attached to PixD or its binding partner, PixE. We observed less cluster formation with monomer-PixD combinations (Supplementary Fig. 1b), indicating that the valency of the module fused to PixD plays a more critical role in cluster formation than that of the module fused to PixE. Additionally, light-induced dissociation was impaired when PixD was conjugated to FTL, which forms a 24-mer. Through our valency optimization (Supplementary Fig. 1), we identified the optimal combination—DsRed-PixD and SNAP$_{tag}$-PixE—for RELISR that resulted in high light-responsiveness and effective condensate formation.

Other optogenetic condensates, such as PixELL[9,16], also utilized the PixD/PixE system. However, their cluster formation depends on the interactions of intrinsically disordered regions (IDRs), resulting in liquid-liquid phase separation (LLPS)[9]. In LLPS environments, the relative freedom of biomolecules enables dynamic and promiscuous interactions within and around IDR-based condensates. In contrast, our RELISR system relies on multivalent interactions between DsRed, PixD, and PixE to form highly immobile clusters in the dark state. Thus, unlike IDR-based optogenetic condensates, our RELISR system enables stable and specific storage of target molecules.

We also directly compared with the optogenetic system, mito-LOVTRAP[48,49]. While RELISR physically isolates cargo in condensates, mito-LOVTRAP anchors Zdk1-fusion proteins to mitochondria in the dark and releases them into the cytosol upon illumination, spatially restricting their function[48]. This approach effectively controls protein localization but does not physically isolate mRNA from the translational machinery. Furthermore, ongoing translation near mitochondria may limit the effectiveness of LOVTRAP-mediated RNA sequestration[52]. These differences highlight a key distinction in sequestration strategy: while LOVTRAP relies on surface tethering, RELISR provides physical exclusion through condensate formation, which is crucial for regulating cargo accessibility.

The light-controlled storage and release of cargo proteins by RELISR was successfully shown in HeLa cells, mouse fibroblasts, and rat primary neurons. We observed differences in cluster size between the soma and neurites of neurons, likely due to the physical constraints unique to each compartment. The local functional output of RELISR may be influenced by the spatial distribution of pre-formed clusters and cargo load, which could constrain response magnitude in certain contexts. Future investigations using the RELISR may help to further explore the local regulation of neuronal proteins, such as neurotransmitter receptors and synaptic proteins, as well as their encoding mRNAs. While the current study does not directly address these aspects, the system's ability to locally regulate protein function (Supplementary Fig. 9) suggests its potential for studying spatially confined biochemical processes relevant to neuronal physiology.

We demonstrated the applicability of Protein-RELISR to a range of proteins with diverse sizes and functions, including GFP (27 kDa), GFP-Vav2 (~72 kDa), and the lipid phosphatase PTEN-GFP (~71 kDa). While the upper limit of cargo size remains undetermined, the inhibition of cargo mRNA translation implies that RELISR system feasible with proteins of diverse sizes. We also note that RELISR cluster properties would vary depending on the cargo characteristics. Our experiments with RELISR clusters targeting GFP, GFP-AD, and mRNA indicate differences in dissociation/re-association kinetics and mobile fractions (Supplementary Fig. 11). GFP nanobody-conjugated PixE could serve as an alternative for multimeric cargo; however, users may need to empirically test constructs to identify the optimal RELISR variant for their specific cargo type.

Our in vivo experiments demonstrate the feasibility of using RELISR for blue light–controlled regulation of mRNA translation in a mice model (Fig. 6). These results provide a foundation for adapting the platform to broader in vivo applications. Although blue light has limited tissue penetration, previous studies have reported effective illumination at depths of up to ~875 μm in internal tissues[50,53,54]. Further developments may incorporate red-shifted optogenetic systems to enhance penetration and minimize phototoxicity[55,56]. While hydrodynamic tail-vein injection enables rapid gene expression, it is not suitable for long-term or tissue-specific delivery. Adeno-associated virus (AAV) vectors, widely used for stable and targeted expression in vivo, offer a more appropriate alternative. The coding sequences of our RELISR constructs fall within the AAV packaging limit[57], making them compatible with this delivery strategy.

In summary, this study introduces RELISR, an optogenetic condensate system capable of reversibly sequestering and releasing diverse proteins and mRNAs in a light-dependent manner. This allows spatiotemporal control over protein function and mRNA translation, with demonstrated applicability across multiple cell types and proof-of-concept in vivo.

Given its ability to dynamically form and dissolve cargo-sequestering compartments in response to light, RELISR offers a modular framework that can be tailored to different biological contexts. While we have demonstrated its functionality using representative proteins and mRNAs across various cell types and in vivo settings, the system is not limited to these examples. Future studies may expand its applicability through cargo-specific engineering, refinement of in vivo delivery methods, or adopting programmable systems such as CRISPR-Cas, thereby enabling precise control over endogenous targets. Through these advances, RELISR has the potential to support a wide range of applications—from dissecting the spatiotemporal dynamics of cellular signaling to uncovering RNA biology mechanisms in physiologically relevant environments.

## Methods

### Ethics statement

Ethical approval for all animal experiments and procedures was obtained from the Institutional Animal Care and Use Committee (IACUC) at the Korea Advanced Institute of Science and Technology (KAIST); the study was conducted under approval number KA2024-100-v1. Six-week-old male BALB/c mice were sourced from Raonbio Service, Yongin-si, South Korea. Pregnant Sprague Dawley rats (E18) were sourced from Seongnam-si, South Korea.

### Plasmid construction

The PixD sequences for the RELISR candidates were amplified from the FUS-FusionRed-PixD plasmid (Addgene plasmid #111503) and cloned into the mCerulean vector between the BspEI and BamHI sites using Gibson assembly. The SNAP$_{tag}$-PixD, mScarlet-PixD, DsRed-PixD, and LanYFP-PixD constructs were generated by swapping fluorescent proteins via AgeI and BsrGI digestion of mCerulean-PixD plasmids. To generate FTL-Citrine-PixD and FTL-SNAP$_{tag}$-PixD constructs, the ferritin (FTL) sequence was amplified from CIB1-MP[58] and inserted between the NheI and AgeI sites of the Citrine-PixD or SNAP$_{tag}$-PixD plasmids.

PixE was amplified from the FUS-Citrine-PixE plasmid (Addgene plasmid #111505) and cloned into the DsRed vector between the BspEI and BamHI sites. The mScarlet-PixE, mCerulean-PixE, and SNAP$_{tag}$-PixE constructs were generated by swapping fluorescent proteins via AgeI and BsrGI digestion of DsRed-PixE plasmid. FTL-Citrine-PixE and FTL-SNAP$_{tag}$-PixE were generated by replacing PixD with PixE via BspEI and BamHI digestion.

To target GFP-conjugated proteins, V$_H$HGFP-DsRed-PixD and V$_H$HGFP-SNAP$_{tag}$-PixE constructs were generated by inserting V$_H$HGFP from V$_H$HGFP-mCherry-CRY2PHR[59] between the NheI and AgeI sites of each plasmid. V$_H$HGFP–V$_H$HGFP–SNAPtag–PixE was constructed by

inserting an amplified $V_H$HGFP–SNAPtag fragment into the $V_H$HGFP–SNAPtag–PixE backbone between the AgeI and BsrGI sites using Gibson assembly.

The EGFP-Vav2(DHPH)-Rit-tail construct was created by excising a PCR-amplified sequence encoding Vav2 (amino acids 167–541) using BspEI and XhoI digestion, inserting it into the pEGFP-C1 vector (Clontech), and then adding the Rit-tail sequence[26] between the EcoRI and BamHI sites. The constructs encoding EGFP-conjugated Rac1(Q61L), RhoG(WT), Tiam1, Cdc42(Q61L), RhoG(Q61L), and PTEN were inserted EcoRI and BamHI sites of pEGFP-C1 vector[59]. iRFP682–PLCδ–PH was generated by replacing the fluorescent protein in GFP–C1–PLCδ–PH (Addgene plasmid #21179) using AgeI and BsrGI digestion.

For mRNA targeting, tdMCP-DsRed-PixD and tdMCP-FusionRed constructs were generated by amplifying tdMCP from phage-ubc-nls-ha-tdMCP-gfp (Addgene plasmid #40649) and inserting the amplified fragment between the NheI and AgeI sites of the DsRed-PixD and FusionRed-C1 vectors. phage-cmv-cfp-24xms2 (indicated as "CFP-MBS x 24" in this study, Addgene plasmid #40651). The Fluc-MBSx24 construct was generated using Gibson assembly by amplifying the Fluc sequence from the pmirGLO vector (E133A, Promega) and inserting it between the AgeI and BsrGI sites of the CFP-MBSx24 plasmid.

For comparison with mitochondrial LOVTRAP, tdMCP–FusionRed–Zdk1 was generated by amplifying Zdk1 from TriEx–mCherry–Zdk1 (Addgene plasmid #81057) and inserting the fragment between the NheI and BsrGI sites of tdMCP–FusionRed (described above). mScarlet–Zdk1–tdMCP was constructed by sequentially amplifying Zdk1 and tdMCP and assembling the fragments using Gibson assembly.

## Cell culture and transfection

HeLa (#CCL-2, ATCC), NIH3T3 (#CRL-1658, ATCC), and HEK293T (#CRL-11268, ATCC) cells were cultured in DMEM (Gibco) supplemented with 10% fetal bovine serum (FBS; Gibco) at 37 °C in a humidified atmosphere containing 10% $CO_2$. The e-Myco™ Mycoplasma PCR Detection Kit (iNtRON) was used routinely to ensure that all cultures were mycoplasma-free. Transfections were carried out using either the Neon transfection system (Invitrogen) or Lipofectamine LTX (Invitrogen) according to the manufacturer's protocols. For NIH3T3 cells, transfection efficiency was optimized by using three pulses at 1,450 V for 10 ms. For live-cell imaging, approximately 6,000–9,000 cells/well were seeded to 96-well plates (μ-Plate 96-well ibiTreat; ibidi GmbH). NIH3T3 cells were initially plated to 6-well plates post-electroporation, allowed to recover overnight, and then transferred to PDL-coated (Sigma) 96-well plates. HEK293T cells were electroporated and then plated to 6-well plates for immunoblot analysis.

## Hippocampal neuron preparation and transfection

Hippocampal neurons were isolated from embryonic day 18 (E18) Sprague Dawley rat embryos. Embryos were transferred into Hank's Balanced Salt Solution (HBSS) supplemented with HEPES, and hippocampi were dissected and enzymatically dissociated in 0.25% trypsin at 37 °C for 15 min, with gentle agitation every 5 min. The tissues were washed three times with HBSS-HEPES supplemented with FBS and mechanically dissociated using fire-polished Pasteur pipettes. The dissociated neurons were filtered through a 70-μm cell strainer (BD Falcon) and plated at 35,000 cells/well in 24-well plates (Cellvis) containing NM10 plating medium comprising 10% horse serum, 2% GlutaMAX, and 1% antibiotic-antimycotic solution (Cytiva HyClone) in Neurobasal Medium (all from Gibco). After 1 h of incubation at 37 °C in a humidified atmosphere with 5% $CO_2$, the NM10 medium was entirely replaced with Neurobasal Medium supplemented with B27. The medium was refreshed every 3 days (replacement of 30%). Neurons were transfected using Lipofectamine LTX following the manufacturer's

protocol. All media and sera were purchased from Gibco, unless otherwise indicated.

## Immunoblot analysis

Whole-cell lysates were prepared under dark conditions using PRO-PREP solution (iNtRON Biotechnology). Proteins (10 μg per sample) were separated by SDS-PAGE on a NuPAGE Novex 4–12% Bis-Tris gel (Invitrogen) and transferred to a nitrocellulose membrane utilizing an iBlot transfer stack and iBlot gel transfer device (Invitrogen) according to the manufacturer's protocol. The membranes were then probed with mouse anti-GFP (1:1000; Abcam, SC-9996) and mouse anti-GAPDH (1:1000; Invitrogen, MA5-15738). After being washed, the membranes were incubated with goat anti-mouse IRDye 680RD secondary antibody (1:15,000; LI-COR Biosciences). All antibodies used in this study have been validated by the manufacturers for western blot in the relevant species and applications. A protein size marker (5 μl, Bio-Rad #161-0373) was used to estimate the molecular weight of each band. The blots were scanned using an Odyssey CLx infrared imaging system (LI-COR Biosciences), and band intensities were quantified with LI-COR Image Studio version 5.5.

## Fluorescent in situ hybridization (FISH)

Stellaris FISH probes targeting MBS sites (as previously designed by the Singer group[47]) or human GAPDH (SMF-2019-1) were labeled with Quasar 670 dye (LGC Biosearch Technologies). Cells in 96-well plates (ibidi) were transfected as described above, incubated for 16 h, fixed in 4% paraformaldehyde solution (Electron Microscopy Sciences) for 15 min at room temperature, and washed three times with 1 × DPBS (Gibco) for 5 min each. The cells were permeabilized by immersion in 70% ethanol (EMD Millipore) for at least 6 h at 4 °C and then washed at room temperature for 5 min with a solution containing 20% Stellaris RNA FISH wash buffer A (LGC Biosearch Technologies), 70% nuclease-free water (Invitrogen), and 10% deionized formamide (Sigma). The cells were then plated and incubated for 16 h in the dark at 37 °C with a 250-nM probe solution prepared in 90% Stellaris RNA FISH hybridization buffer (LGC Biosearch Technologies) and 10% deionized formamide. The cells underwent an additional wash using 20% Stellaris RNA FISH Wash Buffer A, 70% nuclease-free water, and 10% deionized formamide at 37 °C for 30 min in darkness. This was followed by a final rinse with Stellaris RNA FISH Wash Buffer B (LGC Biosearch Technologies) at room temperature for 5 min. Subsequently, a drop of fluorescence mounting medium (Dako) was applied to each well. FISH signals were acquired using a Hamamatsu EM-CCD camera (C9100), with illumination provided by a Nikon INTENSILIGHT C-HGFIE fluorescent lamp. Images were captured with an exposure time of 2 s.

## Live-cell imaging and light stimulation

Live-cell imaging was performed using a Nikon A1R confocal microscope mounted on a Nikon Ti2 platform. Imaging employed a Nikon CFI Plan Achromat VC 60X/1.4 NA objective lens and was controlled via NIS-Elements AR software (64-bit, version 5.20). To maintain physiological condition (37 °C and 10% $CO_2$) during imaging, a Chamlide TC system (Live Cell Instruments) was installed on the microscope stage.

For light stimulation during time-lapse imaging, a 488-nm Cobolt laser integrated into the Nikon photo-stimulation module was used. For whole-cell stimulation, the 488-nm laser was applied at 2.5 μW/cm² intensity across the entire imaged area at 15-s intervals, unless otherwise specified. For moderate light stimulation to analyze dissociation kinetics, the 488-nm laser was applied at 1.3 μW/cm² intensity across the entire imaged area at 30-s intervals for an hour. For LOVTRAP light stimulation experiments, the 457 nm laser was applied at 0.1 μW/cm² intensity across the entire imaged area at 30-s intervals for 30 min.

For repeated light stimulation (3 cycle), 1.3 μW /cm² across the entire imaged area at 10-s intervals. For local stimulation (as shown in Fig. 1), the 488-nm laser was applied at 0.4 μW/cm² intensity to the

region of interest (ROI), labeled as "R1", with 5-s intervals for 45 stimulations. Prior to stimulation, to confirm the RFU of DsRed-PixD signals, images were acquired for 230-s with 5-s intervals under 561-nm laser.

To enable light stimulation for immunoblotting experiments, a custom-built 488-nm LED array was used to ensure uniform blue-light illumination across all wells. Light illumination was performed at an intensity of $4.5\,\mu W/cm^2$ ($90\,\mu W/cm^2$ for LOVTRAP) for 24 h, with a cycle of 5-s on and 2-s off for RELISR experiments, and 50-s on and 10-ec off for LOVTRAP experiments.

Light intensity was measured at the LED pad surface using the photodetector connected to a PM120D power meter (Thorlabs).

## Fluorescence Recovery After Photobleaching (FRAP) experiments

FRAP experiments were conducted using a "bleaching" module and "ND stimulation" tool in Nikon system. To bleach the DsRed signal of Protein-RELISR clusters, a circular region of interest (ROI) was selected inside the target cluster. A total of 306 bleaching pulses (561 nm, $510\,nW/cm^2$) were applied over six cycles, each consisting of 51 pulses over 8 s. Between each bleaching cycle, one image (2-s acquisition) was captured, resulting in a total bleaching session of approximately 60 s.

Following photobleaching, fluorescence recovery was monitored using 561-nm laser excitation ($95\,nW/cm^2$) at 5-s intervals for 5–10 min. For dual-channel imaging, 488-nm laser excitation ($20\,nW/cm^2$) was also used to acquire GFP images and confirm Protein-RELISR cluster targeting of EGFP-tagged cargo.

Fluorescence intensity within the bleached ROI was measured using the "ROI Statistics" tool in Nikon imaging software. Intensities ($I$) were normalized to the average of the first two pre-bleach acquisitions. In all recovery plots, only the final 30 s of the pre-bleach phase are shown for clarity, and the bleaching period itself was omitted from the displayed time axis. Recovery curves were analyzed in GraphPad Prism9 using nonlinear regression with an exponential plateau model, starting from the end of the bleaching session.

Calculation of Mobile fraction: The mobile fraction (M) was calculated using the following formula:

$$Mobile\ fraction(M) = \frac{I_{max} - I_{min}}{I_0 - I_{min}} \quad (1)$$

where $I_0$ is the average fluorescence intensity in the pre-bleach phase, $I_{min}$ is the minimum fluorescence intensity immediately after bleaching, representing the initial drop in intensity, and $I_{max}$ is the maximum fluorescence intensity reached during the post-bleach phase.

This calculation quantifies the fraction of fluorescent molecules that are mobile and able to diffuse into the bleached area, reflecting the dynamic properties of the condensate components.

Determination of Half-Recovery Time ($t_{1/2}$): The half-recovery time ($t_{1/2}$) was defined as the time-point at which the fluorescence intensity reached half of the difference between $I_{min}$ and $I_{max}$. This value was extracted directly from the fitted recovery curves.

## Imaging processing and analysis

Images were aligned and analyzed using Nikon imaging software (NIS-elements AR 64-bit version 5.21). Cell boundaries were manually defined with "Polygon" tool in the "Simple ROI Editor". Fluorescence intensities were quantified using the "ROI Statistics" tool.

Fluorescent clusters were identified using the "Object Count" tool by setting an intensity threshold between 1713 and 4095 arbitrary units (a.u.) and a minimum area of $0.2\,\mu m^2$. For each time point, the sizes and number of clusters per cell were also determined using the "Objective Count" tool. Each cluster was defined as a binary object, and colocalization between fluorescence channels was assessed using the

"Colocalization" tool, with Pearson's correlation coefficient calculated within defined ROI.

Fluorescence intensity profiles were obtained using the "Intensity profiles" analysis tool.

Pseudo-colored images were obtained by selecting the "Green Fire Blue" option in the "Channel Coloring" tool.

Cytosolic intensity was estimated by excluding the contribution of clusters from the total cell signal, using the formula:

$$Cytosolic\ intensity(F) = \frac{(Total\ cell\ intensity - Cluster\ intensity)}{(Total\ Cell\ area - Cluster\ area)}$$

(2)

Dissociation kinetics were analyzed in GraphPad Prism9 using nonlinear regression with one-phase decay or two-phase decay model, following the equation below.

$$One - phase\ decay\ model:\ y = (1 - Plateau) \times e^{-Kx} + Plateau \quad (3)$$

$$Two - phase\ decay\ model:\ y = (SpanFast) \times e^{-K_1 x} + (Spanslow) \times e^{-K_2 x} + Plateau \quad (4)$$

For plateau validation, log-transformed decay curves were analyzed, and mean slopes in the final 20% of the x-axis range were computed to evaluate residual dissociation.

## In vivo light activation and Luminescence imaging

Six-week-old male BALB/c mice were purchased from Raonbio Service (Yongin-si, South Korea) and hydrodynamically transfected by co-injection of tdMCP-FusionRed or mRNA-RELISR and a luciferase reporter plasmid. Transfection was carried out using TransIT-EE Hydrodynamic Delivery Solution (Mirus Bio) with $150\,\mu g$ plasmid DNA per mouse. Prior to injection, abdominal fur was shaved using a chemical depilatory cream. Following transfection, mice were randomly assigned to a dark or blue-light group. The blue-light group was exposed for 16 h to a 473-nm LED light source at $3\,mW/cm^2$, positioned at the bottom of the cage, while the dark group was kept in a cage without blue-light exposure. Bioluminescence imaging was conducted 16 h after transfection. Mice were injected intraperitoneally with $200\,\mu L$ of D-luciferin (15 mg/mL in DPBS; Promega) immediately before imaging. Anesthesia was induced with isoflurane, and imaging was conducted using an IVIS Lumina system (Xenogen) equipped with a CCD camera, using a 1-s exposure and high binning settings. Luminescence signals were quantified using IVIS Lumina software (Xenogen).

## Statistics and reproducibility

All experiments were conducted in at least three biologically independent replicates, with results presented as mean ± standard error of the mean (s.e.m.). GraphPad Prism9 was used for data visualization, graphing and statistical analyses. No statistical method was used to predetermine sample size. Sample sizes were chosen based on commonly accepted standards in the field and prior experience with similar experimental systems. No data were excluded from the analyses. Cells and mice were randomized to experimental groups. Blinding was not applicable, as all replicates were uniformly prepared and analyzed by the same researchers. All micrographs in this study are representative images of experiments carried out with at least three repetitions.

## Reporting summary

Further information on research design is available in the Nature Portfolio Reporting Summary linked to this article.

## Data availability

All data supporting the findings of this study are available within the main manuscript and Supplementary Information. Raw confocal microscopy images are not publicly available due to large file size limitations and data format compatibility, but can be obtained from the corresponding author upon reasonable request. These requests will be evaluated and fulfilled within 5 working days. Source data are provided with this paper.

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

## Acknowledgments
We thank all of the members of the Heo laboratory for their support and advice. We thank Dr. Na Yeon Kim and Dr. Nury Kim for helpful discussions during the early stages of this project. This work was supported by the Samsung Science and Technology Foundation under Project No. SSTF-BA1902-06. This work was also supported by the Bio&Medical Technology Development Program of the National Research Foundation (NRF) funded by the Korean government (MSIT) (No. RS-2023-00263628).

## Author contributions
C.L. and W.D.H. conceived the project and directed the work. C.L., Jeonghye Yu, J.S., D.Y., and W.D.H designed the experiments. C.L., Jeonghye Yu, J.S., D.Y., and Y.H performed the experiments. C.L., M.L., and Y.P. supported the data analysis. C.L., Jihwan Yu, and W.D.H. wrote the manuscript. All authors discussed the data and contributed to the manuscript.

## Competing interests
The authors declare no competing interests.
