## [Transparent Peer Review file · Nature Communications]

Optogenetic Storage and Release of Protein and mRNA in Live Cells and Animals

Corresponding Author: Professor Won Do Heo

Version 0:

Reviewer comments:

Reviewer #1

(Remarks to the Author)

In this manuscript, Lee et al. describe the development of a novel optogenetic platform that enables the light-controlled storage and release of proteins and mRNAs. First, the authors constructed a basic framework that self-assembles and can dissociate in response to light, based on the PixD-PixE system, following careful optimization. They demonstrated that the system achieves a satisfactory cluster formation rate and exhibits the light responsiveness necessary for optogenetic control.

Next, the authors showed that light can manipulate GFP, Vav2 and other proteins in the Rho signaling pathway. Finally, they applied this approach to regulate mRNA storage and translation in cultured cells and mice.

While I believe the authors have successfully shown that optogenetic control of proteins and mRNA is feasible—primarily using fluorescent proteins and their corresponding mRNAs—the manuscript falls short in demonstrating the broader biological and medical relevance of this technique. The authors present only a single example where cell morphology was influenced by manipulating Rho pathway proteins using this approach. Although I recognize the novelty of the technique, I am uncertain whether it constitutes a breakthrough (Line 348) in biological research.

I would recommend considering this manuscript for publication in Nature Communications, provided that the authors further demonstrate how this technique offers significant insights into the local regulation of protein function and translation, as emphasized in lines 334–337, and show its broader implications for cell physiology.

Reviewer #2

(Remarks to the Author)

In the present manuscript, the authors report on the selective storage and release of proteins and mRNAs using an optogenetically controlled condensate platform, termed RELISR (Reversible Light-Induced Store and Release). Consisting of the blue-light responsive protein pair PixD/PixE, multivalent proteins and cargo-binding domains, RELISR stores a specific cargo in biomolecular condensates in the dark. Upon blue-light illumination, the condensates dissociate and release their cargo molecules into the cellular environment. The authors demonstrated this reversible mechanism for various protein cargoes in primary neurons and fibroblasts and for mRNAs in vitro and in mice and were able to control cellular processes using the RELISR system.

The study was carefully designed and presented in an understandable manner. The hypotheses were tested in an appropriate manner and the conclusions drawn from the experiments are justified and supported by a variety of experiments with appropriate controls.

The manuscript addresses a topic of interest to a broad community by providing insight into light-controlled condensate formation, cellular signaling processes, protein function and mRNA translation. In addition, it adds to the still small pool of studies using the PixD/PixE optogenetic switch.

However, the following points should be addressed prior to publication.

Major points:

- While the authors have demonstrated the reversibility of their system, it should be tested whether there is a limit to its reversibility. By subjecting the system to cycles of dark and light exposure, repeated storage and complete release of cargo

and thus a potential reversibility limit could be easily tested.

- The clusters of both the Protein-RELISR and the mRNA-RELISR systems appear to dissolve at different rates. The Toettcher group recently published an optogenetic tool to dissolve biomolecular condensates by light-dependent recruitment of the highly soluble maltose-binding protein (MBP) into the condensates (Brumbaugh-Reed et al., 2024, Nature Comm., DOI: 10.1038/s41467-024-50858-0). Therefore, it would therefore be interesting to see how the cargo affects the condensates in terms of stiffness, dissolution times and reversibility. A comparative experiment to test these parameters in one cell line with the previously used cargoes and a highly soluble protein such as MBP should therefore be performed. In addition, a discussion of the differences between the two systems would be beneficial.

- Please indicate in the materials and methods section whether the fluorescence intensity data from the FRAP experiments were normalized before calculating the mobile fraction.

- In my opinion, the *in vivo* experiment performed in this study does not provide major or novel insight. Therefore, please demonstrate the efficacy and application of the Protein-RELISR system with functional proteins and their effect in mice, if such experiments have been performed already. Alternatively, please discuss in detail the points to consider and the problems and limitations that may arise when implementing this system in mice.

Minor points:

Main text

- Experienced optogeneticists may know that the PixD/PixE switch is the only dissociation switch that also allows heterooligomerization, which is the central principle of this system. But less experienced readers might wonder why such a rarely used switch was chosen. A sentence on this topic might clarify the reasoning behind the choice of the optogenetic tool.

- Line 94: Supple Fig 1e is referenced, but it should be 1c

- Line 97: Supple Fig 1e is referenced, is this correct?

- Chapter I: Figure 1h and I are not referenced in the text

- Chapter III: Figures 3i and 3k are not referenced in the text

Figures

- Figure 1: The title is not ideal, as one might expect a description of the acronym RELISR after the colon. Something like "RELISR constitutes a light-dissociable container using multivalent proteins" would clarify it.

- Figure 2b figure legend: The names of the two systems are not used consistently, the shorter name shown in front of the parentheses for the first system is missing for the second. Compare: VHHGFP-DsRed-PixD (VHHGFP-DsRed-PixD)/SNAPtag-PixE (magenta) and VHHGFP-SNAPtag-PixE (VHHGFP-SNAPtag-PixE)/DsRed-PixD (blue).

- Figure 3c: Adding GFP colocalization either in the figure legend or adding "GFP" before the y-axis label "colocalization" would improve comprehensibility.

- Figure 3g and h: The blue boxes around the panels are a bit too thin to be visible. While the blue numbers above indicate the illumination quite well on their own, making the box a bit thicker would make it even easier to see and understand.

- Figure 4b: Merge panel, "Protein-RELISR" is shown in the panel, but according to the figure legend and main text it should be only "RELISR"

- Figure 4c: Yellow arrows referred to in the figure legend are missing

- Figure 4g figure legend: The formula for the fold change calculation seems to be reversed.

- Please add legends/ descriptions for the supplementary movies

Materials and Methods

- Plasmid construction: please homogenize the references to the Addgene plasmids, some are referenced by number and link, some by link only, and one by number, link and RRID, the first two having "RRID" in parentheses with the actual RRID number missing.

Reviewer #3

(Remarks to the Author)

Reviewer #4

(Remarks to the Author)

PixD PixE mechanism should be described, including known reversibility

In the manuscript "Optogenetic storage and release of protein and mRNA in live cells and animals", Lee et al develop RELISR as a method to bind protein or RNA cargoes into large assemblies of fusion proteins of PixD and PixE for baseline sequestration and light-mediated release. They first adapted PixD-PixE to generate large assemblies by screening for fusions with multimeric proteins, with the fusion of tetrameric DsRed and PixD achieving large assemblies that can be dissolved by blue light. Cargoes of GFP-tagged proteins of interest were then sequestered by fusing a GFP-recognizing Vhh domain to DsRed-PixD, while cargoes of MBS-containing RNA were sequestered by fusing the MBS-binding tdMCP to DsRed-PixD. The authors demonstrate the ability of the assemblies to inhibit function of GFP-tagged proteins and to release functional proteins upon illumination in non-neuronal and neuronal cells, with reversal in the dark occurring over minutes. They demonstrate the ability to suppress translation of RNA cargoes by assembly formation (to 27% of control), and the ability to activate it by light (to 73% of control). Most impressively, they introduce the system into mouse liver and achieve

8.5-fold induction of protein production with light exposure.

Overall the manuscript is well written and presented. The method is potentially powerful and widely applicable, especially for synthetic systems where GFP-Vhh and MBS-tdMCP can be used as generalizable interactions. The ability to control protein translation with light is especially novel and could find wide use in basic research. There are only 3 questions that I think need to be addressed before publication (only 2 of which may require experimentation).

1. The Vhh-DsRed-PixD fusion even after disassembly has a valency of 4 in the Vhh (it could be more if PixD dimerization is preserved, see below). This could create light-independent aggregates if the bound target protein is also a multimer. Thus the demonstrated system using Vhh-DsRed-PixD + monomeric PixE may not be as applicable as one using DsRed-PixD + Vhh-PixE for multimeric targets. I believe the authors should either demonstrate that the current system works fine on a multimeric target, or demonstrate that functional proteins can be released using a DsRed-PixD + Vhh-PixE system, even if a higher-affinity Vhh must be used, so that users can select that configuration for multimeric target proteins. Note a tandem Vhh may be a substitute for a higher-affinity Vhh.

The suggestion of using a higher affinity Vhh derives from the observation that Vhh-DsRed-PixD + PixE was better at sequestering GFP than DsRed-PixD + Vhh-PixE. This is actually expected as the unaltered stoichiometry of PixD:PixE is believed to be 10:4 before alteration and the ratio might increase further with the tetramerization of PixD by DsRed. So fusing Vhh to DsRed-PixD likely creates more Vhh copies in assemblies than fusing to PixE would, which shifts the equilibrium toward binding by LeChatelier's principle and also may generate more avidity. Thus it is not surprising that Vhh-DsRed-PixD was better at sequestering GFP than Vhh-PixE. However, a well sequestering DsRed-PixD + Vhh-PixE might be possible by using higher-affinity Vhh or doubling the Vhh.

2. It is important to state the energies used as irradiance (watts per area) rather than just watts. Thus any time a W figure is cited (e.g. lines 111-116, but many other places too), it should be restated as the calculated or measured W per area. Also it is not clear in line 129 what is meant by 230 seconds at 5 second intervals. How many seconds was light on and how many seconds was light off, and of course what was the irradiance?

3. Mitochondrial LOVTRAP shares the same ability as RELISR in sequestering a fusion protein and releasing it after illumination. It has also been used successfully in neuritic processes (<https://doi.org/10.1038/s41467-017-00044-2>). Thus the unique aspect of RELISR appears to be RNA sequestration and release. It should be straightforward for the authors to test whether a LOVTRAP fusion to the same tdMCP they used in RELISR can effectively sequester RNA away from being translated.

Addressing the first two points will fully describe the method and guarantee that it can work on multimeric proteins, and thus avoid having readers perform unnecessary experiments. Addressing the last point will allow readers to know if RELISR is unique in RNA sequestration.

Minor points:

1. The authors should mention the mw of PixD and PixE. The fact that PixD is the photoactive domain and that the chromophore is FAD should also be mentioned.

2. The authors should explicitly describe how their system is different from PixELLS, which also used PixD and PixE to make condensates, and they should cite Dine et al 2018 Cell Systems.

3. The authors should discuss how their system is similar and different from LOVTRAP.

4. After dissociation, PixD is believed to remain dimeric while PixE is a monomer (Dine et al 2018 Cell Systems). Can the authors speculate as to why are the assemblies with DsRed-PixD able to dissolve completely, then? One would expect DsRed-PixD even when dissociated to create networks via the 4+2 valencies of DsRed and PixD. Maybe the PixD dimerization doesn't happen in the DsRed fusion, but perhaps the authors can explain with an atomic model.

5. The experiment of Figure 2J demonstrates the condensates are not phase separated liquid-liquid droplets but are more gel-like. Line 68 should perhaps state that multivalent interactions facilitate condensate formation (which is a broader term) rather than phase separation.

Version 1:

Reviewer comments:

Reviewer #1

(Remarks to the Author)

In the revised manuscript, the authors explored local release of PTEN and Inp54p, demonstrating that this lipid phosphatase pathway can be modulated using the RELISR system. These additional results further support the conclusions of the study.

While Supplementary Figure 9 is informative, I personally find the Inp54p data more compelling. I would like to leave it to the authors' discretion whether to include the results for PTEN, Inp54p, or both in the Supplementary Figure 9.

Reviewer #2

(Remarks to the Author)

The authors have adequately addressed the comments and suggestions from my previous review.

In my initial review, I suggested a more extensive exploration of the repeated cargo release over multiple light cycles, a comparative analysis using a highly soluble cargo such as MBP, and an expanded discussion on potential steps toward clinical implementation of the RELISR system.

The authors have submitted a thorough and thoughtfully revised version of their manuscript. They have performed extensive additional experiments, made impactful changes in response to mine, as well as the other reviewers' comments, and have taken great care to address the comments and suggestions provided.

The revisions have improved the quality of the manuscript, which in my opinion, is now ready to be accepted for publication.

Reviewer #3

(Remarks to the Author)

Reviewer #4

(Remarks to the Author)

The authors have addressed all my questions well and in my opinion have also addressed the other reviewers' questions satisfactorily. I thus recommend the manuscript for publication.

**Reviewer #1 (Remarks to the Author):**

In this manuscript, Lee et al. describe the development of a novel optogenetic platform that enables
the light-controlled storage and release of proteins and mRNAs. First, the authors constructed a basic
framework that self-assembles and can dissociate in response to light, based on the PixD-PixE system,
following careful optimization. They demonstrated that the system achieves a satisfactory cluster
formation rate and exhibits the light responsiveness necessary for optogenetic control.

Next, the authors showed that light can manipulate GFP, Vav2 and other proteins in the Rho signaling
pathway. Finally, they applied this approach to regulate mRNA storage and translation in cultured cells
and mice.

While I believe the authors have successfully shown that optogenetic control of proteins and mRNA is
feasible—primarily using fluorescent proteins and their corresponding mRNAs—the manuscript falls
short in demonstrating the broader biological and medical relevance of this technique. The authors
present only a single example where cell morphology was influenced by manipulating Rho pathway
proteins using this approach. Although I recognize the novelty of the technique, I am uncertain
whether it constitutes a breakthrough (Line 348) in biological research.

I would recommend considering this manuscript for publication in Nature Communications, provided
that the authors further demonstrate how this technique offers significant insights into the local
regulation of protein function and translation, as emphasized in lines 334–337, and show its broader
implications for cell physiology.

We sincerely appreciate the reviewer's constructive feedback and acknowledge the need to
further demonstrate the broader biological relevance of RELISR beyond fluorescent protein regulation
and Rho pathway signaling. In response, we have conducted additional experiments and revised the
manuscript accordingly (Line 283-294, Line 449-455, and Supplementary Fig. 9).

**Fig R1.** Schematic illustration of PTEN- and Inp54p-mediated dephosphorylation of PIP₃ and
PIP₂ at the plasma membrane.

To address concerns regarding the broader applicability and local regulatory potential of our
platform, we have expanded our study to include PTEN and Inp54p as candidate targets in neuronal
contexts. These proteins were selected due to their established roles in phosphoinositide signaling,
critical for neuronal function. Specifically, PTEN converts PIP₃ to PIP₂, negatively regulating the
PI3K/Akt pathway involved in neuronal growth and synaptic plasticity, while Inp54p converts PIP₂
to PI4P, affecting ion channel activity and actin dynamics. To monitor real-time changes in PIP₂ levels, we
used the PH domain of PLCδ as a biosensor, which selectively binds to PIP₂ at the plasma membrane.

To demonstrate the feasibility of local release of functional protein, we co-expressed PTEN-GFP,
 VHHGFP–DsRed–PixD, SNAPtag–PixE, and iRFP682–PLC δ –PH domain in primary hippocampal
 neurons (Supplementary Fig. 9). PTEN-GFP showed colocalization with VHHGFP–DsRed–PixD clusters
 under dark conditions ($r = 0.7587$; Supplementary Fig. 9c). Upon local blue-light stimulation at the
 defined ROI in neurites (488 nm, $0.4 \mu\text{W}/\text{cm}^2$, 45 pulses with 5-s intervals for 10 mins), we observed
 the decrease in cytosolic PLC δ –PH intensity (fold change = 0.8314), accompanied by an increase at the
 membrane region (fold change = 1.3543; $P = 0.0205$; Supplementary Fig. 9f). In contrast, the cytosolic
 intensity of the unstimulated regions exhibited significantly less intensity change ($P < 0.0001$;
 Supplementary Fig. 9d), confirming the precise conversion of PI(3,4,5)P $_3$ to PI(4,5)P $_2$ at the
 illuminated sites, consistent with PTEN's known function.

 **Figure R2.** Local release of Inp54p using Protein-RELISR in neuronal substructure.

 We next tested whether local release of Inp54p alters the subcellular distribution of PLC δ –PH
 domain. To ensure membrane targeting, a Rit-tail sequence was fused to the C-terminus of Inp54p,
 generating EGFP–Inp54p–Rit-tail. This construct was co-expressed with Protein-RELISR and
 iRFP682–PLC δ –PH domain in rat hippocampal neurons. The stimulation ROI was defined on the
 neurite (blue box in Fig. R2b) and illuminated using the same conditions as for PTEN release. Local
 light stimulation triggered Inp54p release, resulting in PLC δ –PH domain accumulation at the
 stimulated site (Fig. R2b). We observed increased fluorescence intensity near the membrane and in
 the adjacent cytosol as confirmed by time-lapse imaging (fold change = 1.319; Fig. R2c) and intensity
 profile analysis (Fig. R2d). In contrast, non-stimulated regions showed a modest decrease in PLC δ –PH
 domain signal, consistent with redistribution toward the illuminated site (fold change = 0.8432; Fig.
 R2c). These results indicate that RELISR enables localized Inp54p activity and demonstrate its
 applicability to neuronal signaling studies.

We acknowledge potential challenges associated with the local functional application of RELISR.
 For instance, light-induced release is primarily confined to regions where clusters are pre-formed,
 which could influence spatial flexibility depending on the application. Additionally, the extent of
 functional output is influenced by amount of protein stored within each cluster, which in some contexts
 may limit the magnitude of the biological response. Nevertheless, RELISR offers a broadly applicable
 platform for spatiotemporal release of proteins, supporting precise and modular manipulation of
 cellular functions.

Since our current study does not provide direct evidence for the local regulation of mRNA
 translation in neurons, we have revised the discussion in Lines 449–455 to reflect this limitation more
 accurately:

*“The local functional output of RELISR may be influenced by the spatial distribution of pre-formed*
 *clusters and cargo load, which could constrain response magnitude in certain contexts. Future*
 *investigations using the RELISR system may help to further explore the local regulation of neuronal*
 *proteins, such as neurotransmitter receptors and synaptic proteins, as well as their encoding mRNAs.*
 *While the current study does not directly address these aspects, the system’s ability to locally regulate*
 *protein function suggests its potential for studying spatially confined biochemical processes relevant to*

*neuronal physiology.”*

Finally, we highlight that RELISR enables precise spatiotemporal control of signaling pathways at
both whole-cell and subcellular levels. As demonstrated, the localized release of PTEN and Inp54p
induced redistribution of the PLC δ PH domain. Although we did not directly assess neuronal activity,
these results illustrate the system’s potential for dissecting subcellular signaling dynamics with high
spatial resolution.

We have also clarified how RELISR conceptually differs from previously reported systems such as
PIXELL (Lines 430–436), OptoMBP (Reviewer 2, major comment 2), and LOVTRAP (Lines 365–399),
as requested by the reviewers. Unlike these existing tools, RELISR uniquely enables optogenetic
control of mRNA translation and has demonstrated in vivo applicability.

We thank Reviewer #1 once again for their thoughtful and constructive feedback, which helped
93 us to strengthen the biological relevance and broader applicability of the RELISR system.

**Reviewer #2 (Remarks to the Author):**

In the present manuscript, the authors report on the selective storage and release of proteins and
mRNAs using an optogenetically controlled condensate platform, termed RELISR (Reversible Light-
Induced Store and Release). Consisting of the blue-light responsive protein pair PixD/PixE,
multivalent proteins and cargo-binding domains, RELISR stores a specific cargo in biomolecular
condensates in the dark. Upon blue-light illumination, the condensates dissociate and release their
cargo molecules into the cellular environment. The authors demonstrated this reversible mechanism
for various protein cargoes in primary neurons and fibroblasts and for mRNAs in vitro and in mice
and were able to control cellular processes using the RELISR system.

The study was carefully designed and presented in an understandable manner. The hypotheses were
tested in an appropriate manner and the conclusions drawn from the experiments are justified and
supported by a variety of experiments with appropriate controls.

The manuscript addresses a topic of interest to a broad community by providing insight into light-
controlled condensate formation, cellular signaling processes, protein function and mRNA
translation. In addition, it adds to the still small pool of studies using the PixD/PixE optogenetic
switch.

However, the following points should be addressed prior to publication.

**Major points:**

2.1. While the authors have demonstrated the reversibility of their system, it should be tested
whether there is a limit to its reversibility. By subjecting the system to cycles of dark and light
exposure, repeated storage and complete release of cargo and thus a potential reversibility limit
could be easily tested.

We thank Reviewer 2 for the insightful comment regarding the potential limits of system
reversibility. To address this point, we conducted additional experiments to evaluate whether Protein-
RELISR can repeatedly store and release cargo over multiple cycles of light stimulation and recovery
(Supplementary Fig. 4).

HeLa cells expressing Protein-RELISR (VHHGFP–DsRed–PixD and SNAP-tag–PixE pairs) along
with GFP were exposed to blue light concurrently during GFP imaging (488 nm, 1.3 $\mu\text{W}/\text{cm}^2$; 10-
second intervals for 10 minutes), followed by a 20-minute dark recovery period (Supplementary Fig.
4a). To assess the extent of cargo release, we quantified the cytosolic intensities of VHHGFP–DsRed–
PixD and GFP at each time point during image acquisition (Supplementary Fig. 4b). Cytosolic intensity
was estimated by excluding the contribution of clusters from the total cell signal, using the formula:
(total cell intensity – cluster intensity) / (total cell area – cluster area).

As expected, during each light period, both the cytosolic DsRed intensity (Protein-RELISR) and
the cytosolic GFP intensity consistently increased (DsRed: $F_{10}/F_0 = 1.307$, $F_{40}/F_0 = 1.232$, $F_{70}/F_0 = 1.297$;
GFP: $F_{10}/F_0 = 1.164$, $F_{40}/F_0 = 1.167$, $F_{70}/F_0 = 1.219$). These elevations indicate effective cargo release in
all three rounds (Supplementary Fig. 4b).

During the dark recovery period, we observed a gradual reduction in both signals (DsRed: F_{30}/F_0
= 1.081, $F_{60}/F_0 = 1.159$; GFP: $F_{30}/F_0 = 1.072$, $F_{60}/F_0 = 1.134$), reflecting partial reassembly of
condensates and re-sequestration of cargo. Notably, although the re-sequestration efficiency declined
slightly over successive cycles, Protein-RELISR retained its responsiveness to light. In the third cycle,
the cytosolic DsRed signal increased upon stimulation ($F_{70}/F_0 = 1.297$) and subsequently decreased
during recovery ($F_{90}/F_0 = 1.121$), indicating that the system remains functionally reversible under
repeated illumination (Supplementary Fig. 4b).

These new results have been included in the main text (Lines 179–186), Supplementary Figure 4,
and experimental details are described in the relevant Methods section.

2.2. The clusters of both the Protein-RELISR and the mRNA-RELISR systems appear to dissolve at
 different rates. The Toettcher group recently published an optogenetic tool to dissolve biomolecular
 condensates by light-dependent recruitment of the highly soluble maltose-binding protein (MBP)
 into the condensates (Brumbaugh-Reed et al., 2024, Nature Comm., DOI: 10.1038/s41467-024-
 50858-0). Therefore, it would therefore be interesting to see how the cargo affects the condensates
 in terms of stiffness, dissolution times and reversibility. A comparative experiment to test these
 parameters in one cell line with the previously used cargoes and a highly soluble protein such as
 MBP should therefore be performed. In addition, a discussion of the differences between the two
 systems would be beneficial.

 We thank the reviewer for highlighting the recent OptoMBP strategy (Brumbaugh-Reed et al.,
 2024) and for suggesting a comparative analysis of how cargo properties influence condensate
 behavior. In response, we conducted additional experiments using MBP as a cargo in the Protein-
 RELISR system and directly compared its behavior to GFP and mRNA cargoes. These comparisons
 included colocalization, light-induced dissociation kinetics, re-association dynamics, and FRAP-based
 mobility analyses.

 **Figure R3.** Cargo-dependent colocalization and light-induced dissociation of RELISR clusters

To first assess whether our tool can stably store MBP within Protein-RELISR, we fused MBP to the
 N-terminus of GFP, as previous studies have shown that N-terminal tagging enhances solubility of
 fused protein (Sachdev and Chirgwin 1998). Co-expression with Protein-RELISR allowed us to observe
 clear colocalization of MBP-GFP with Protein-RELISR (DsRed) under dark conditions (Fig. R3a, $r =$
 0.7401). This colocalization was comparable to that observed with GFP alone ($r = 0.8130$) and mRNA
 (CFP-MBS; $r = 0.8332$) (Fig. R3a, one-way ANOVA, n.s. $P > 0.1$). Although GFP clusters appeared larger,
 quantitative analysis revealed no significant cargo-dependent differences in the initial cluster size (Fig.
 R3b; one-way ANOVA, n.s., $P > 0.5$).

To examine potential differences in dissociation kinetics, we applied moderate light stimulation
 (488nm, $1.3 \mu\text{W}/\text{cm}^2$, with 30-seconds interval for an hour) and quantified the relative decrease in
 cluster area over time (Fig. R3c). Using a one-phase decay model, we estimated the dissociation
 kinetics of clusters containing different cargoes. The rate constant K , which reflects the speed of cargo
 release (higher values indicate faster dissociation), was highest for mRNA ($K = 16.26$), followed by GFP
 ($K = 10.52$) and MBP ($K = 3.517$). After 1 hour of light stimulation, each plateau value of fitted-curve
 indicated that GFP clusters exhibited the most complete dissociation ($A_1/A_0 = 0.2786$), whereas MBP
 and mRNA clusters retained approximately half of their initial area (MBP: $A_1/A_0 = 0.4687$; mRNA-
 RELISR: $A_1/A_0 = 0.4960$). However, log-scale transformation revealed that the MBP signal, continued
 to decline gradually with a measurable slope (≈ -0.053 in \log_{10} scale). In contrast, the slopes for mRNA
 and GFP clusters were near zero ($\approx -1.2 \times 10^{-6}$ and -4.8×10^{-4} , respectively), consistent with
 stabilization at a plateau. These results suggest that while mRNA-RELISR initiates dissociation rapidly,
 a fraction of the clusters persists over time. GFP-containing clusters displayed the most efficient
 dissociation in terms of both rate and extent, whereas MBP-containing clusters exhibited slower, more
 gradual dissociation kinetics (Fig. R3c).

Figure R4. Cargo-dependent reversibility of RELISR clusters under repeated light stimulation

As described in our response to Reviewer 2's major comment 1, we assessed the reversibility of Protein-RELISR by targeting EGFP through three successive cycles of light stimulation and dark recovery (Supplementary Fig. 4 and Fig. R4a). To extend this analysis, we conducted the same experiment under identical conditions using Protein-RELISR targeting MBP-GFP and mRNA-RELISR co-expressed with CFP-MBS (Fig R4b, and R4c).

When targeting MBP-GFP (Fig R4b), the cytosolic DsRed signal increased upon each light stimulation ($F_{10}/F_0 = 2.207$, $F_{40}/F_0 = 1.812$, $F_{70}/F_0 = 2.049$), indicating light-induced cluster disassembly. However, following the dark recovery periods, the cytosolic DsRed intensity did not markedly decrease except after the first cycle ($F_{30}/F_0 = 1.886$, $F_{60}/F_0 = 1.950$, $F_{110}/F_0 = 2.149$), suggesting limited re-association of MBP-GFP into RELISR clusters compared to the EGFP group. The cytosolic MBP-GFP signal did not show a detectable decline ($F_{10}/F_0 = 1.647$, $F_{30}/F_0 = 1.796$, $F_{40}/F_0 = 1.831$, $F_{60}/F_0 = 1.895$, $F_{70}/F_0 = 2.113$) indicating slower recovery kinetics of MBP targeting clusters.

In contrast, the mRNA-RELISR system exhibited more clear light-dependent behavior (Fig R4c). Light stimulation led to a substantial increase in cytosolic DsRed intensity ($F_{10}/F_0 = 2.791$, $F_{40}/F_0 = 2.993$, $F_{90}/F_0 = 3.284$), while subsequent dark periods resulted in a noticeable decrease ($F_{30}/F_0 = 2.026$, $F_{60}/F_0 = 2.290$, $F_{110}/F_0 = 2.284$), consistent with light-induced disassembly and dark-mediated reassembly of the clusters. These results suggest that different cargoes affect the reversibility kinetics of RELISR clusters, with MBP-GFP inducing slower cluster recovery compared to GFP and mRNA.

PixD is known to require time (Masuda et al. 2004) to return to its multimerization-prone dark-state conformation following photo-dissociation. During this recovery period, scaffold interactions may remain transient or incomplete. Thus, in the case of EGFP and mRNA, the partial re-association may be explained by insufficient time for PixD to fully revert to its dark-state conformation (Fig R4a and c). In contrast, the markedly different reassembly pattern observed with MBP (Fig R4b) suggests an additional inhibitory factor beyond limited PixD recovery. We hypothesize that MBP, due to its large hydrophilic and electrostatically active surface, disrupts reassembly by sterically or electrostatically hindering either PixD-PixD interactions, PixD-PixE interactions, or both. In this context, the inhibitory influence of MBP may outweigh the partially restored assembly forces of PixD, resulting in reduced reassembly efficiency.

**Figure R5.** Cargo-dependent mobility of RELISR clusters assessed by FRAP (GFP, MBP, and
mRNA cargos)

To further investigate how cargo influences the mobility of RELISR clusters, we conducted
fluorescence recovery after photobleaching (FRAP) experiments (Fig. 2j-l, Fig. R5a-c, and
Supplementary Fig. 11). We analyzed the mobile fraction and half-recovery time from FRAP
experiments, as described in the Materials and Methods section.

Clusters storing MBP showed a lower mobile fraction than those storing GFP, although this
difference did not reach statistical significance (MBP: 0.09444; GFP: 0.2027; $P = 0.0611$, one-way
ANOVA; Fig. R5d). mRNA-RELISR exhibited the lowest mobile fraction (0.0577), which was
significantly different from GFP ($P = 0.013$) but not from MBP ($P = 0.6708$, one-way ANOVA). Recovery
half-times did not significantly differ across cargo types ($P > 0.16$, one-way ANOVA), though GFP
clusters exhibited slower and more variable recovery (GFP: 127.5 s; MBP: 56.59 s; mRNA-RELISR:
55.94 s; Fig. R5e).

Although MBP is commonly employed to enhance protein solubility and reduce aggregation, its
behavior as condensate cargo is notably context-dependent. Our findings using the Protein-RELISR
system demonstrate that cargo identity, such as MBP incorporation, significantly influences molecular
dynamics within condensates, despite minimal effects on initial cluster size. Specifically, MBP-GFP
stably integrates into clusters via a high-affinity GFP nanobody, showing comparable initial
localization to GFP alone (Fig. R3a). However, MBP-containing condensates exhibit notably slower
dissociation kinetics upon light stimulation (Fig. R3c), reduced re-association after repeated
illumination cycles (Fig. R4). These behaviors collectively suggest that MBP incorporation effected
condensate network, likely due to its larger molecular size and potential steric interactions with the
PixD-based scaffold.

Interestingly, MBP's stabilizing effect on RELISR condensates contrasts sharply with observations
from the OptoMBP system (Brumbaugh-Reed et al., 2024). In OptoMBP, MBP recruitment to pre-
formed FUS-IDR condensates via light-induced interaction promotes condensate dissolution,
presumably by disrupting scaffold interactions through the introduction of solubilizing domains.
Although the exact molecular mechanism remains speculative in that context, the distinct behaviors
highlight that MBP's impact depends strongly on the temporal context and mode of its incorporation.
Unlike OptoMBP, in RELISR, MBP is pre-integrated into the scaffold structure, reinforcing rather than
disrupting condensate stability.

Yet, despite MBP's stabilizing role within RELISR condensates, its presence paradoxically impairs
dynamic reassembly following photodissociation, possibly due to interference with scaffold recovery.
Thus, while cargo identity such as MBP critically modulates physical and kinetic properties of
synthetic condensates, its precise functional outcome—whether stabilizing or destabilizing—is
context-sensitive. Our results underscore the necessity for careful consideration of cargo identity,
temporal context to predict and control behavior in synthetic condensate systems such as RELISR.

These newly generated data and analyses have been incorporated into the revised manuscript
(Line 331-343) and supplementary figure 4 and 11. We believe these additions directly address the
reviewer's comments and substantially enhance the mechanistic insight into how cargo properties
influence condensate behavior in the RELISR system.

**2.3.** Please indicate in the materials and methods section whether the fluorescence intensity data
from the FRAP experiments were normalized before calculating the mobile fraction.

We thank the reviewer 2 for the careful comment. We have revised the *Materials and Methods*
section to clarify that the fluorescence intensity values were normalized to the average of the first two
pre-bleach acquisitions prior to calculating the mobile fraction and half-recovery time.

In addition to this clarification, we also updated the description of the FRAP procedure to include
more detailed bleaching parameters, image acquisition settings, and the curve-fitting method used in
the recovery analysis.

2.4. In my opinion, the *in vivo* experiment performed in this study does not provide major or novel
insight. Therefore, please demonstrate the efficacy and application of the Protein-RELISR system
with functional proteins and their effect in mice, if such experiments have been performed already.
Alternatively, please discuss in detail the points to consider and the problems and limitations that
may arise when implementing this system in mice.

We agree with Reviewer 2's precise comment that our *in vivo* demonstration does not provide
major or novel insights. Our demonstration using hydrodynamic tail-vein injection was to validate that
the mRNA-RELISR platform is feasible *in vivo*, proof-of-concept for its function in a mammalian
context.

We acknowledge inherent limitations arising from our reliance on PixD, a protein containing a
blue-light-sensitive flavin domain. Additionally, we discussed potential future delivery strategies, such
as adeno-associated virus (AAV)-based systems, extending beyond our current proof-of-concept
demonstration.

An inherent limitation of the current system is the reliance on blue light for activation, which
exhibits limited tissue penetration. Nevertheless, recent studies have demonstrated that blue-light-
responsive optogenetic systems can function effectively *in vivo* under specific illumination conditions.
For instance, Li et al. (2022) and Morikawa et al. (2020) reported blue light penetration depths of up
to 750–875 μm in tissues such as the liver and kidney using LED plate-based illumination. Similarly,
Yu et al. (2024) demonstrated successful genome editing *in vivo* using blue-light photoactivation. To
extend RELISR's utility to deeper tissues and complex systems, future iterations may incorporate red-
shifted optogenetic modules (e.g., Qiao et al., 2024; Leopold et al., 2022), which offer improved tissue
penetration and reduced phototoxicity.

Another limitation of our *in vivo* demonstration lies in the gene delivery method. We employed
hydrodynamic tail-vein injection, which enables rapid and transient transgene expression, typically
peaking within 8–24 hours post-injection (according to the manufacturer's instructions; Mirus Bio,
TransIT-EE). While suitable for initial validation, long-term and tissue-specific expression will likely
require viral delivery. Notably, the coding sequences of RELISR (3060 bp), Protein-RELISR (3426 bp),
and mRNA-RELISR (3783 bp) are well within the AAV packaging limit (~ 4.7 kb including regulatory
elements), making AAV-based delivery feasible for future applications.

Additional factors such as protein stability, cargo degradation, clearance kinetics, and potential
immunogenicity also warrant careful consideration in future *in vivo* adaptations.

We thank the reviewer for this important suggestion and have revised the manuscript accordingly
to clarify the intent of our *in vivo* experiments and expand on technical limitations, possible
improvements, and broader applicability (Lines 464–472).

**Minor points:**

**Main text**

(6) -Experienced optogeneticists may know that the PixD/PixE switch is the only dissociation switch
that also allows heterooligomerization, which is the central principle of this system. But less
experienced readers might wonder why such a rarely used switch was chosen. A sentence on this topic
might clarify the reasoning behind the choice of the optogenetic tool.

We thank the reviewer for highlighting this important point. In response, we have added a brief
explanation in Lines 75–81 outlining the unique features of the PixD/PixE system and our rationale

for adopting it in the design of RELISR.

-Line 94: Supple Fig 1e is referenced, but it should be 1c

The reviewer is correct, and we have now revised the relevant panel accordingly (Line 105).

-Line 97: Supple Fig 1e is referenced, is this correct?

We thank the reviewer for pointing this out. The correct reference is to Supplementary Figure 1d,
and we have revised the text accordingly (Line 109).

-Chapter I: Figure 1h and I are not referenced in the text

We thank reviewer for pointing this out. We have now added references to Fig. 1h and 1i in
the text (Line 140).

-Chapter III: Figures 3i and 3k are not referenced in the text

The reviewer is correct. We have now cited Fig. 3i and Fig. 3k in the appropriate locations in the
text (Lines 237).

**Figures**

-Figure 1: The title is not ideal, as one might expect a description of the acronym RELISR after the colon.
Something like "RELISR constitutes a light-dissociable container using multivalent proteins" would
clarify it.

We thank the reviewer for the kind suggestion and fully agree with the comment. We have now
revised the figure 1 title to: "RELISR constitutes a light-dissociable container using multivalent
proteins," as proposed.

-Figure 2b figure legend: The names of the two systems are not used consistently, the shorter name
shown in front of the paratheses for the first system is missing for the second. Compare: VHHGFP-
DsRed-PixD (VHHGFP-DsRed-PixD)/SNAPtag-PixE (magenta) and VHHGFP-SNAPtag-PixE (VHHGFP-
SNAPtag-PixE)/DsRed-PixD (blue).

We thank the reviewer for kindly pointing this out. We have revised the legend to ensure
consistent use of system names and abbreviations across both constructs: VHHGFP-DsRed-PixD
(VHHGFP-PixD) / SNAPtag-PixE (magenta) and VHHGFP-SNAPtag-PixE (VHHGFP-PixE) / DsRed-PixD
(blue)

-Figure 3c: Adding GFP colocalization either in the figure legend or adding "GFP" before the y-axis label
"colocalization" would improve comprehensibility.

We thank the reviewer for the helpful suggestion. We have now revised all relevant y-axis labels
to read "Colocalization with (E)GFP" to improve clarity and consistency.

-Figure 3g and h: The blue boxes around the panels are a bit too thin to be visible. While the blue
numbers above indicate the illumination quite well on their own, making the box a bit thicker would
make it even easier to see and understand.

We thank the reviewer for the helpful suggestion. We have increased the thickness of the blue
box outlines in Fig. 3g and 3h to enhance visibility and improve clarity.

-Figure 4b: Merge panel, “Protein-RELISR” is shown in the panel, but according to the figure legend
and main text it should be only “RELISR”

The reviewer is correct, and we thank them for pointing out this oversight. We have now corrected
the label in the panel of Fig. 4b from “Protein-RELISR” to “RELISR” to match the figure legend and main
text.

-Figure 4c: Yellow arrows referred to in the figure legend are missing

We thank the reviewer for pointing this out. We have removed the mention of the yellow arrows
from the figure legend to reflect the current panel content.

-Figure 4g figure legend: The formula for the fold change calculation seems to be reversed.

The reviewer is correct—the formula for fold change was incorrectly stated. We have now
corrected it in the figure legend. Thank you for pointing this out.

**-Please add legends/ descriptions for the supplementary movies**

We apologize for the oversight regarding these details. We have now added detailed descriptions
for all supplementary movies, which can be found in the revised Supplementary Information.

**Materials and Methods**

-Plasmid construction: please homogenize the references to the Addgene plasmids, some are
referenced by number and link, some by link only, and one by number, link and RRID, the first two
having “RRID” in parentheses with the actual RRID number missing.

We thank the reviewer for the precise observation. We have now corrected the formatting
inconsistencies and added the missing RRID numbers to ensure uniform referencing of Addgene
plasmids.

Overall, we have carefully addressed Reviewer 2’s detailed and well-organized minor comments. All
corresponding changes have been highlighted in yellow throughout the manuscript. We sincerely
thank the reviewer once again for their thoughtful feedback.

**Reviewer #3 (Remarks to the Author):**

I co-reviewed this manuscript with one of the reviewers who provided the listed reports. This is part
of the Nature Communications initiative to facilitate training in peer review and to provide
appropriate recognition for Early Career Researchers who co-review manuscripts.

We appreciate your time and thoughtful evaluation of our manuscript. We also fully support the
initiative to involve and recognize Early Career Researchers in the peer review process and commend
the efforts to facilitate their training in this important area.

**Reviewer #4 (Remarks to the Author):**

PixD PixE mechanism should be described, including known reversibility

In the manuscript “Optogenetic storage and release of protein and mRNA in live cells and animals”,
Lee et al develop RELISR as a method to bind protein or RNA cargos into large assemblies of fusion
proteins of PixD and PixE for baseline sequestration and light-mediated release. They first adapted
PixD-PixE to generate large assemblies by screening for fusions with multimeric proteins, with the
fusion of tetrameric DsRed and PixD achieving large assemblies that can be dissolved by blue light.
Cargoes of GFP-tagged proteins of interest were then sequestered by fusing a GFP-recognizing Vhh
domain to DsRed-PixD, while cargoes of MBS-containing RNA were sequestered by fusing the MBS-
binding tdMCP to DsRed-PixD. The authors demonstrate the ability of the assemblies to inhibit
function of GFP-tagged proteins and to release functional proteins upon illumination in non-
neuronal and neuronal cells, with reversal in the dark occurring over minutes. They demonstrate the
ability to suppress translation of RNA cargoes by assembly formation (to 27% of control), and the
ability to activate it by light (to 73% of control). Most impressively, they introduce the system into
mouse liver and achieve 8.5-fold induction of protein production with light exposure.

Overall the manuscript is well written and presented. The method is potentially powerful and widely
applicable, especially for synthetic systems where GFP-Vhh and MBS-tdMCP can be used as
generalizable interactions. The ability to control protein translation with light is especially novel and
could find wide use in basic research. There are only 3 questions that I think need to be addressed
before publication (only 2 of which may require experimentation).

**1.** The Vhh-DsRed-PixD fusion even after disassembly has a valency of 4 in the Vhh (it could be more
if PixD dimerization is preserved, see below). This could create light-independent aggregates if the
bound target protein is also a multimer. Thus the demonstrated system using Vhh-DsRed-PixD +
monomeric PixE may not be as applicable as one using DsRed-PixD + Vhh-PixE for multimeric
targets. I believe the authors should either demonstrate that the current system works fine on a
multimeric target, or demonstrate that functional proteins can be released using a DsRed-PixD +
Vhh-PixE system, even if a higher-affinity Vhh must be used, so that users can select that
configuration for multimeric target proteins. Note a tandem Vhh may be a substitute for a higher-
affinity Vhh.

The suggestion of using a higher affinity Vhh derives from the observation that Vhh-DsRed-PixD +
PixE was better at sequestering GFP than DsRed-PixD + Vhh-PixE. This is actually expected as the
unaltered stoichiometry of PixD:PixE is believed to be 10:4 before alteration and the ratio might
increase further with the tetramerization of PixD by DsRed. So fusing Vhh to DsRed-PixD likely
creates more Vhh copies in assemblies than fusing to PixE would, which shifts the equilibrium
toward binding by LeChatelier’s principle and also may generate more avidity. Thus it is not
surprising that Vhh-DsRed-PixD was better at sequestering GFP than Vhh-PixE. However, a well
sequestering DsRed-PixD + Vhh-PixE might be possible by using higher-affinity Vhh or doubling the
Vhh.

We greatly appreciate Reviewer 4’s insightful suggestions and critical feedback. As accurately
noted, DsRed-PixD is inherently multimeric due to PixD oligomerization, and this multimerization is
further enhanced by DsRed-induced tetramerization. According to Le Chatelier’s principle, this
significantly shifts the equilibrium toward cluster formation. We agree that, particularly when
targeting multimeric cargo, the enhanced multivalency of DsRed-PixD may stabilize light-independent
assemblies that resist dissociation. To address this, as suggested by the reviewer, we generated an
alternative construct by fusing a tandem GFP nanobody to PixE (VHHx2-PixE).

We compared three Protein-RELISR constructs—VHHGFP-DsRed-PixD/SNAP_{tag}-PixE (VHH-
PixD), VHHGFP-PixE/DsRed-PixD (VHH-PixE), and tandem GFP nanobody-PixE (VHHx2-PixE)—
using both monomeric GFP (Supplementary Fig. 5a) and multimeric GFP-AD (dodecamer) cargo
proteins (Supplementary Fig. 5b).

We first assessed colocalization between EGFP and each construct by calculating Pearson's
correlation coefficients (r). VHH-PixD exhibited the strongest colocalization ($r = 0.8397$), followed by
VHH-PixE ($r = 0.7309$) and VHHx2-PixE ($r = 0.6918$). The differences between VHH-PixD and both
VHH-PixE (* $P = 0.0375$) and VHHx2-PixE (** $P = 0.0013$) were statistically significant, whereas the
difference between VHH-PixE and VHHx2-PixE was not (n.s., $P = 0.5386$, two-way ANOVA;
Supplementary Fig. 5c). These results suggest that VHH-PixD provides superior sequestration
efficiency for monomeric cargo.

To assess EGFP release dynamics, we illuminated cells expressing each construct with EGFP for 1
463 hour using a 488 nm laser ($1.3 \mu\text{W}/\text{cm}^2$) at 30-second intervals (Supplementary Fig. 5d, e). Cluster
area was quantified at each time point, normalized to the initial value, and fitted using nonlinear decay
models (one- or two-phase decay; see Methods). VHH-PixD displayed the fastest release kinetics ($K =$
10.52), followed by VHHx2-PixE ($K = 5.498$) and VHH-PixE ($K = 2.031$), supporting the conclusion
that VHH-PixD is optimal for EGFP requiring both efficient sequestration and rapid release. While
VHH-PixE showed slower dissociation, VHHx2-PixE exhibited improved kinetics, implying that
tandem nanobody fusion alters the assembly or accessibility of PixE within condensates. This
structural modification may influence both condensate formation and light-induced release behavior.

When EGFP-AD (dodecamer) was targeted (Supplementary Fig. 5b), Pearson's correlation
coefficients between EGFP-AD and GFP nanobody-conjugated constructs remained consistently high
across all variants (Supplementary Fig. 5c). VHHx2-PixE showed the highest colocalization ($r = 0.9060$),
followed by VHH-PixD ($r = 0.8797$) and VHH-PixE ($r = 0.8508$); however, these differences were not
statistically significant (all $P > 0.2$, two-way ANOVA; Supplementary Fig. 5c). Dissociation kinetics for
EGFP-AD revealed notable differences (Supplementary Fig. 5e): VHHx2-PixE exhibited the fastest
initial dissociation ($K_1 = 348.1$), and VHH-PixE also showed biphasic decay kinetics ($K_1 = 81.37$). In
contrast, VHH-PixD showed the slowest dissociation ($K = 7.075$), with $\sim 89\%$ of clusters persisting
after prolonged illumination. This supports the reviewer's concern that VHH-PixD may form 'light-
independent aggregates' when the presence of multimeric cargo further shifts the equilibrium toward
aggregation.

Therefore, when targeting multimeric proteins, VHHx2-PixE or VHH-PixE may be advantageous
over VHH-PixD. However, the tandem nanobody in VHHx2-PixE does not inherently enhance targeting
efficiency, as shown by the monomeric EGFP results (Supplementary Fig. 5a, c). Therefore, construct
selection should be tailored to the oligomeric state and properties of the cargo to ensure optimal
sequestration and release performance.

We have now incorporated all relevant data and discussions into the main text (Lines 187-214
and Lines 459-463), supplementary figure 5 with corresponding legends, and the relevant Materials
and Methods section.

**2.** It is important to state the energies used as irradiance (watts per area) rather than just watts.
Thus any time a W figure is cited (e.g. lines 111-116, but many other places too), it should be
restated as the calculated or measured W per area. Also it is not clear in line 129 what is meant by
230 seconds at 5 second intervals. How many seconds was light on and how many seconds was light
off, and of course what was the irradiance?

We appreciate the reviewer's important comment regarding the completeness of the light
stimulation parameters. In response, we have updated the manuscript to include detailed light
conditions—including exact power ($\mu\text{W}/\text{cm}^2$), interval, pulse duration, and total stimulation time—
for each experiment. These specifications are now provided in the figure legends or main text
corresponding to each panel, and also summarized in the "Live-cell imaging and light stimulation"
section of the Materials and Methods (Line 580-603). We thank the reviewer for helping us improve
the clarity and precision of the manuscript.

**3. Mitochondrial LOVTRAP shares the same ability as RELISR in sequestering a fusion protein and**
**releasing it after illumination. It has also been used successfully in neuritic processes**
**(<https://doi.org/10.1038/s41467-017-00044-2>).** Thus the unique aspect of RELISR appears to be
RNA sequestration and release. It should be straightforward for the authors to test whether a
LOVTRAP fusion to the same tdMCP they used in RELISR can effectively sequester RNA away from
being translated.

We sincerely thank Reviewer 4 for suggesting a comparison between our RNA sequestration
system (mRNA-RELISR) with a LOVTRAP-based approach using tdMCP. Mitochondrial LOVTRAP
shares conceptual similarity with RELISR in restricting fusion protein activity in the dark state and
releasing it upon illumination. Given LOVTRAP's established role in controlling protein localization,
we evaluated whether fusing tdMCP to LOVTRAP could achieve similar RNA sequestration and
translational control.

We co-expressed Tom20-mVenus-LOV2 (Tom20-LOV2) with either tdMCP-FusionRed-Zdk1
(tdMCP-Zdk1) or mScarlet-Zdk1-tdMCP (Zdk1-tdMCP) in HeLa cells, examining their localization and
light responsiveness (Supplementary Fig. 13a, b). In darkness, both tdMCP-Zdk1 and Zdk1-tdMCP
colocalized with mitochondrial LOV2. Upon blue-light illumination (457 nm, 0.1 $\mu\text{W}/\text{cm}^2$, 30-second
intervals for 30 min), both constructs dispersed into the cytoplasm and subsequently relocalized to
the mitochondria under dark conditions, confirming light-dependent localization regardless of fusion
orientation (Supplementary Fig. 13a, b).

We then investigated whether LOVTRAP-based systems control mRNA localization via
fluorescence in situ hybridization (FISH) of CFP-MBS transcripts (Supplementary Fig. 13c-e). HeLa
cells co-expressing Tom20-LOV2, CFP-MBS, and either tdMCP-Zdk1 or Zdk1-tdMCP underwent
patterned blue-light illumination (LED, 0.09 mW/cm^2 , 50 s ON/10 s OFF) or remained in darkness for
24 hours. After fixation and FISH labeling (Quasar 670), MBS FISH signals exhibited moderate
colocalization with mitochondria under dark conditions (tdMCP-Zdk1: $r = 0.6482$; Zdk1-tdMCP: $r =$
0.6222 ; $P = 0.9982$, one-way ANOVA; Supplementary Fig. 13e). Upon light exposure, the colocalization
coefficients slightly decreased (tdMCP-Zdk1: $r = 0.5248$; Zdk1-tdMCP: $r = 0.5175$), though the changes
were not statistically significant ($P = 0.0767$ and 0.0960 , respectively, one-way ANOVA; Supplementary
Fig. 13e).

Next, we assessed whether LOVTRAP-based systems could regulate mRNA translation in a light-
dependent manner (Supplementary Fig. 13f, g). Western blot analysis revealed negligible differences
in CFP protein levels between dark and light conditions for tdMCP-Zdk1 (fold change [light/dark] =
0.981), Zdk1-tdMCP (1.020), and the tdMCP-FusionRed control (1.053). These results indicate that
mitochondrial LOVTRAP systems fused to tdMCP do not effectively regulate cargo mRNA translation
under the tested conditions.

Together, these results demonstrate that mitochondrial LOVTRAP does not robustly control RNA
localization or translation. Mechanistically, mitochondrial LOVTRAP anchors Zdk1 fusion proteins to
the mitochondrial surface in the dark, thereby restricting their spatial distribution. While the system
can localize mRNA to mitochondria, it does not physically isolate it from the translational machinery.
Moreover, ongoing translation near mitochondria may further limit the effectiveness of LOVTRAP-
mediated RNA sequestration (Lesnik, Golani-Armon, and Arava 2015). In contrast, RELISR forms
protein-RNA condensates via multivalent interactions, physically sequestering mRNA, resulting in
robust translational suppression in the dark and efficient activation upon light-induced disassembly.

We appreciate the reviewer's suggestion, which prompted a careful evaluation of the LOVTRAP
system for RNA translational control. Through these comparative experiments, we clarified the
mechanistic distinctions between LOVTRAP and RELISR, demonstrating that mRNA-RELISR enables
more effective light-dependent RNA sequestration and translational regulation. These findings, along
with all relevant data and analyses, have now been incorporated into the main text (Line 365-399 and

Line 437-444), Supplementary Figure 13 with its legend, and the Materials and Methods section. We
believe these insights highlight the novelty of the RELISR platform and may inform the future
development of optogenetic tools for RNA biology.

Addressing the first two points will fully describe the method and guarantee that it can work on
multimeric proteins, and thus avoid having readers perform unnecessary experiments. Addressing
the last point will allow readers to know if RELISR is unique in RNA sequestration.

**Minor points:**

1. The authors should mention the mw of PixD and PixE. The fact that PixD is the photoactive domain
and that the chromophore is FAD should also be mentioned.

We have added additional information regarding PixD and PixE, including their molecular weights,
the presence of an FAD chromophore in PixD, its classification as a BLUF-domain photoreceptor, and
their hetero-oligomerization properties (Line 72-81). We appreciate the suggestion, which helped
improve the clarity and completeness of the manuscript.

2. The authors should explicitly describe how their system is different from PixELLS, which also used
PixD and PixE to make condensates, and they should cite Dine et al 2018 Cell Systems.

We have now clarified the distinction between our system and the previously reported PixELL
platform more explicitly in the Discussion section (Line 430-436). In addition to the existing reference
to Zhao et al., 2019 (Nat. Chem. Biol.), we have also included the relevant citation of Dine et al., 2018
(Cell Systems) as suggested. We appreciate this helpful comment, which allowed us to present a more
comprehensive comparison and improve the overall clarity of the manuscript.

3. The authors should discuss how their system is similar and different from LOVTRAP.

In response to the reviewer's insightful suggestion, we have added a discussion clarifying the
mechanistic similarities and differences between our system and LOVTRAP (Line 437-444). We thank
the reviewer for this valuable feedback, which helped us strengthen the conceptual positioning of our
platform within the broader landscape of optogenetic tools.

4. After dissociation, PixD is believed to remain dimeric while PixE is a monomer (Dine et al 2018
Cell Systems). Can the authors speculate as to why are the assemblies with DsRed-PixD able to
dissolve completely, then? One would expect DsRed-PixD even when dissociated to create networks
via the 4+2 valencies of DsRed and PixD. Maybe the PixD dimerization doesn't happen in the DsRed
fusion, but perhaps the authors can explain with an atomic model.

Fig R6. HeLa cell expressing DsRed-PixD (Scale bar 10µm)

We greatly appreciate this insightful and fundamental question. To address whether DsRed-PixD
alone can sustain clusters after dissociation, we expressed DsRed-PixD without PixE and found that it
failed to form visible condensates under dark conditions (Fig. R6). This observation suggests that PixE

interaction is essential for cluster formation, and that light-induced disruption of the PixD–PixE
complex is a critical trigger for dissociation. Therefore, we speculate that although PixD may remain
dimeric after light exposure, the absence of multivalent hetero-oligomeric interactions with PixE
renders DsRed-PixD unable to maintain stable assemblies on its own. While we did not conduct
atomic-level modeling in this study, we believe our experimental results provide a reasonable
mechanistic explanation for the observed complete cluster dissolution.

5. The experiment of Figure 2J demonstrates the condensates are not phase separated liquid-liquid
droplets but are more gel-like. Line 68 should perhaps state that multivalent interactions facilitate
condensate formation (which is a broader term) rather than phase separation.

We agree with the reviewer’s comment and have revised the wording in Line 70 to use
“condensate formation” instead of “phase separation,” as suggested. Thank you for pointing this out
and helping us improve the clarity of the manuscript.

We thank Reviewer 4 once again for the opportunity to improve our manuscript, particularly with
respect to the optimization for multimeric cargo, the clarification of novelty, and the inclusion of more
precise details.

**Reviewer #1 (Remarks to the Author):**

In the revised manuscript, the authors explored local release of PTEN and Inp54p, demonstrating that
this lipid phosphatase pathway can be modulated using the RELISR system. These additional results
further support the conclusions of the study. While Supplementary Figure 9 is informative, I personally
find the Inp54p data more compelling. I would like to leave it to the authors' discretion whether to
include the results for PTEN, Inp54p, or both in the Supplementary Figure 9.

We thank the reviewer for the thoughtful comments and are pleased that the additional data were
found informative. After careful discussion, we have decided to retain the current format of
Supplementary Figure 9 without further changes, and we appreciate the reviewer's flexibility in
leaving this choice to our discretion.

**Reviewer #2 (Remarks to the Author):**

The authors have adequately addressed the comments and suggestions from my previous review. In
my initial review, I suggested a more extensive exploration of the repeated cargo release over multiple
light cycles, a comparative analysis using a highly soluble cargo such as MBP, and an expanded
discussion on potential steps toward clinical implementation of the RELISR system. The authors have
submitted a thorough and thoughtfully revised version of their manuscript. They have performed
extensive additional experiments, made impactful changes in response to mine, as well as the other
reviewers' comments, and have taken great care to address the comments and suggestions provided.
The revisions have improved the quality of the manuscript, which in my opinion, is now ready to be
accepted for publication.

We sincerely appreciate the reviewer's encouraging feedback. We are glad that the additional
experiments and revisions have strengthened the manuscript, and we are grateful for the valuable
suggestions that guided these improvements.

**Reviewer #3 (Remarks to the Author):**

I co-reviewed this manuscript with one of the reviewers who provided the listed reports. This is part
of the Nature Communications initiative to facilitate training in peer review and to provide appropriate
recognition for Early Career Researchers who co-review manuscripts.

We thank the reviewer and acknowledge the Nature Communications initiative supporting early-
career researcher participation in the peer-review process.

**Reviewer #4 (Remarks to the Author):**

The authors have addressed all my questions well and in my opinion have also addressed the other
reviewers' questions satisfactorily. I thus recommend the manuscript for publication.

We are grateful for the reviewer's recommendation and appreciate the constructive comments that
have greatly contributed to the improvement of our manuscript.